# Understanding and Controlling a Maze-solving Policy Network

## Abstract

To understand the goals and goal representations of AI systems, we carefully study a pre-trained reinforcement learning policy that solves mazes by navigating to a range of target squares, similarly to how model organisms are studied in biology. We find this network pursues multiple context-dependent goals, and we further identify circuits within the network that correspond to one of these goals. In particular, we identified eleven channels that track the location of the goal. By modifying these channels, either with hand-designed interventions or by combining forward passes, we can partially control the policy. We show that this network contains redundant, distributed, and retargetable goal representations, shedding light on the nature of goal-direction in trained policy networks.

## 1 Introduction

To safely deploy AI systems, we need to be able to predict their behavior. Traditionally, researchers do so by evaluating how a model behaves across a range of inputs—for example, with model-written evaluations (Perez et al., 2022), or on static benchmark datasets (Hendrycks et al., 2020; Lin et al., 2021; Liang et al., 2022). Moreover, practitioners usually align AI systems by specifying good behavior, such as via expert demonstrations or preference learning (e.g., Christiano et al., 2017; Hussein et al., 2017; Ouyang et al., 2022; Glaese et al., 2022; Touvron et al., 2023).

However, behavioral analysis and control methods can be misleading. In particular, models may *appear* to be aligned with human goals but competently pursue unintended or even harmful goals when deployed. This behavior is known as *goal misgeneralization* and has been demonstrated by Shah et al. (2022) and Langosco et al. (2023). Moreover, it may be dangerous (Ngo, 2022).

In this work, we therefore investigate the internal objectives (i.e., goals) of trained systems. Intuitively, if we understand the goals of a system, we can better predict the system's behavior in novel contexts during deployment. We focus on goals, while AI interpretability (e.g., Elhage et al., 2021; Fan et al., 2021; Zhang et al., 2021) often pursues a more general understanding of different models.

In particular, we investigate a maze-solving reinforcement learning policy network trained by Langosco et al. (2023); our work is thus analogous to how model organisms are studied in biology (Ankeny & Leonelli, 2020), a paradigm which is also emerging within AI research (Hubinger et al., 2023; 2024). This network exhibits goal misgeneralization—it sometimes ignores a given maze's cheese square in favor of navigating to the top-right corner, which is where the cheese was placed during training (fig. 1a). Moreover, because the policy operates in a human-understandable environment, we can easily interpret its actions and underlying goals. Altogether, this network thus represents an interesting case study.

First, we demonstrate *the trained policy network pursues multiple, context-dependent goals* (§2.1). In ∼5,000 mazes, we examine the policy's choices at *decision squares*—maze locations where the policy must choose between the cheese (the intended generalization) and the historical location of cheese during training (misgeneralization). By using a few features of each maze, we can predict whether the policy network misgeneralizes. This predictability suggests the policy pursues different goals depending on certain maze conditions.

We then find internal representations of these goals. *We identify eleven residual channels that track the location of the cheese* (fig. 1b; §2.2). We demonstrate that these channels primarily affect the behavior of the policy through the location of the cheese, rather than other maze factors. This shows there are circuits in the trained policy network that track this goal. To our knowledge, we are the first to pinpoint internal goal representations in a trained policy network.

We corroborate these findings by showing we can *steer the policy without additional training* (§3). We modify the activations either through manual hand-designed edits to the eleven channels, or by combining the activations corresponding to forward passes. By doing so, we change the policy's behavior in predictable ways. Instead of updating the network, we steer the network by interacting with its "internal motivational API."

Overall, our research clarifies the internal goals and mechanisms in pretrained policy networks. We find that these systems have a nuanced and context-dependent set of goals that can be partially understood and even controlled through activation engineering approaches.

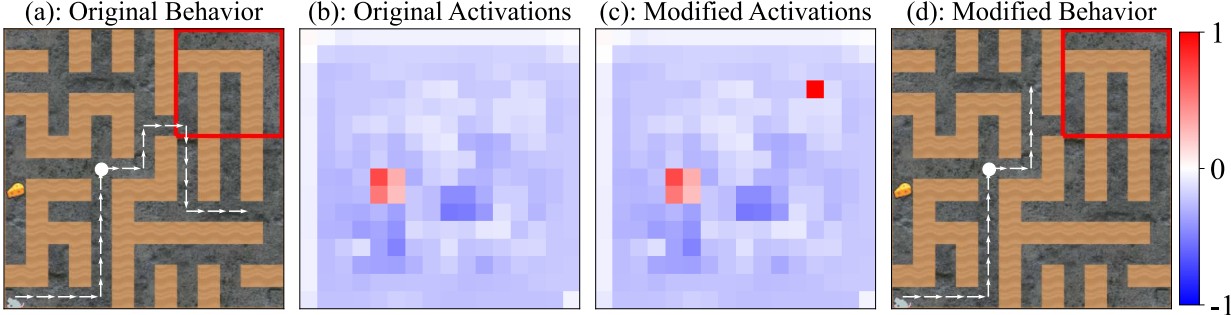

Figure 1: **Understanding and controlling a maze-solving policy.** (a) We examine a maze-solving policy network that navigates within a maze towards a goal location, marked by cheese. During training, the cheese was placed in the upper right $5 \times 5$ corner of the maze—*the historical goal location.* However, during deployment, the cheese may be placed anywhere. The white dot shows a *decision square* where the policy must choose between navigating to the cheese and the top-right corner. (b) We identify residual channels whose activations track the location of the cheese. (c) We manually set one of these activations to $+5.5$. (d) We retarget the policy. Due to the modified activation during the forward pass, the policy goes to the location implied by the edited activation.

## 2 Understanding the Maze-solving Policy Network

We study a maze-solving policy network trained by Langosco et al. (2023). The network is deep, with 3.5M parameters and 15 convolutional layers—see appendix A. The network solves mazes to reach a goal: the cheese. But it exhibits *goal misgeneralization*. It sometimes capably pursues *an unintended goal* at deployment. In this case, the policy often navigates towards the top-right corner (where the cheese *was* placed during training) rather than to the actual cheese (fig. 1a).

During training, the cheese is placed within the top right $5{\times}5$ corner of each randomly generated maze. During deployment, the cheese may be anywhere. The mazes are procedurally generated using the Procgen benchmark (Cobbe et al., 2020). We also consider other policy networks which were pretrained with different historical cheese regions.

We chose this network because it exhibits goal misgeneralization. Furthermore, the network is large enough to be challenging for humans to understand. Finally, the maze environment is easy to visualise, and policies in this environment can be easily understood as making spatial tradeoffs.

**Section overview.** We focus on understanding the goals and goal representations of the maze-solving policy network. First, we examine whether we can predict the generalization behavior of the network by

performing a statistical analysis of the factors that affect the policy's behavior (§2.1). Following this, we identify several residual channels within the network that track the location of the cheese (§2.2). We find the network pursues multiple context-dependent goals, and these goals are internally represented in redundant, distributed ways.

## 2.1 Understanding The Maze-Solving Policy Through Behavioral Statistics

In this environment, the training algorithm does not produce a policy that consistently navigates to the cheese (fig. 2). Specifically, in some mazes, the policy navigates to the cheese, but in other mazes, the *same* policy navigates to the historical cheese location.[1] This suggests the network has the capability to pursue at least two distinct objectives: (i) navigating to the cheese; and (ii) navigating to the top-right corner. We now examine whether behavior can be predicted based on environmental factors. If environmental factors are predictive of the goal pursued by the network, this suggests the goal selected by the network to pursue is context-dependent, rather than chosen at random.

| Maze A | Maze B | Maze C | Maze D |

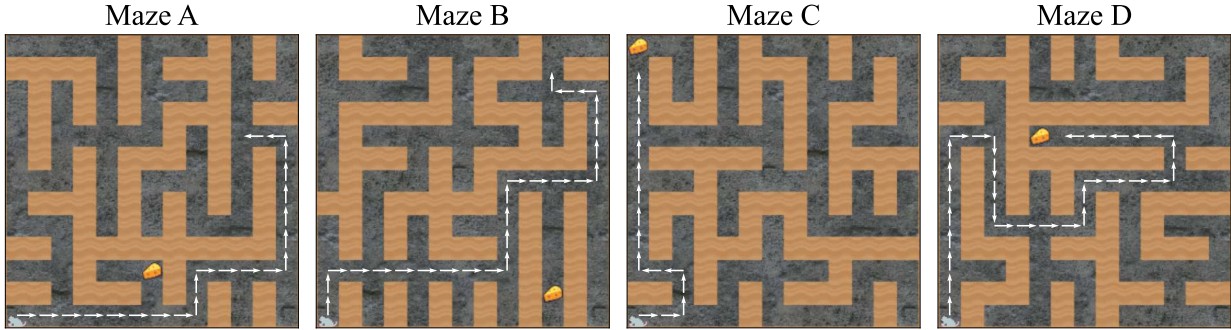

Figure 2: **The policy network pursues multiple goals.** During training, the cheese was always in the top right corner of the maze. We show trajectories in four mazes not from the training distribution. In mazes A and B, the policy ignores the cheese and navigates to the historical goal location (the top-right corner). However, in mazes C and D, the agent navigates to the cheese.

**Experiment details.** We now examine whether we can predict whether the policy navigates to the cheese or the historical cheese location based on maze factors. To do so, we considered 5K mazes where the policy must choose between these goals at a *decision square* (marked by white dots in fig. 1; see also fig. 14 in the appendix). We conducted 10 iterations of train/validation splitting with a validation size of 20%. In each iteration, we performed $\ell_1$-regularized logistic regression to predict whether a network navigates to the cheese for a given environment.

We hypothesized several different environmental factors that may affect the policy's behavior. However, we run our primary analysis only with the following features, which had robust effects across the different analyses: (i) the Euclidean distance from the top-right corner to the cheese; (ii) the step distance from the decision square to the cheese; and (iii) the Euclidean distance from the decision square to the cheese. See appendix B for further details and illustration of these features.

**Results.** Logistic regression on these features achieves an average accuracy of 82.4%, substantially exceeding the 71.4% accuracy of always predicting "reaches cheese." Our three maze features provide substantial information about the goal the policy pursues, which is evidence that the policy pursues *context-dependent* goals. As explored more thoroughly in appendix B, the Euclidean distance from the decision square to the cheese predicts the network's behavior, even after controlling for the step distance from the decision square to the cheese.[2] This indicates that the network's goal pursuit is *perceptually activated* by visual proximity to cheese.

---

[1]In certain mazes (such as fig. 1), the policy doesn't navigate to the cheese *or* to the top-right corner.

[2]These findings are mostly consistent across over a dozen different policy networks trained with different historical cheese locations (see appendix B).

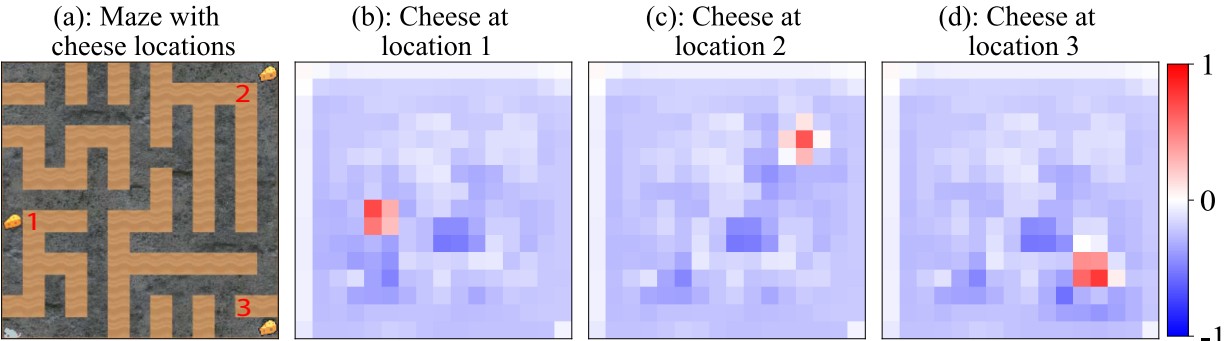

Figure 3: **Network channels track the goal location.** We show the activations for channel 55 after the first residual block of the second IMPALA block. The activations of channel 55 are a $16 \times 16$ grid. We plot the activation values for the same maze when the cheese is placed in different locations. (b-d) show that channel 55 tracks the cheese. See appendix E.1 for more examples.

## 2.2 Finding Goal-Motivation Circuits in the Maze-Solving Policy Network

We have seen that the policy network pursues multiple, context-dependent goals. The network likely contains circuits that correspond to these goals. We identify circuits for the goal of navigating towards the cheese location. Specifically, we find eleven channels about halfway through the network that track the location of the cheese. We consider the network activations after the the first residual block of the second IMPALA block (see fig. 11 in the appendix). At this point of the forward pass, there are 128 separate $16 \times 16$ channels, meaning there are 32,768 activations.

First, we find that some of these channels track the location of the cheese. fig. 3 shows the activations of channel 55 for mazes where the goal is placed in different locations (further examples in appendix E.1). The positive activations (marked in red) correspond to the location of the cheese. By visual inspection, we found that 11 out of these 128 channels track the cheese, showing that the goal representation is redundant. We refer to these 11 channels as the "cheese-tracking" channels.

Suppose these "cheese-tracking channels" do, in fact, track the cheese. Then if we resample their activations (Chan et al., 2022) from another maze with the cheese in the same location,[3] this resampling should not affect the behavior of the network. Moreover, if we resample these activations from a maze where the cheese is placed in a different location, the network should behave as if the cheese were placed in that location. We now test this hypothesis.

First, we visually investigate the effect of resampling the activations of the cheese tracking channels from different mazes (fig. 4; more examples in appendix E.2). Indeed, resampling the activations of these "cheese-tracking" channels modifies the network of the behavior *if* the activations were sampled from another maze where the cheese is in a different location. In contrast, resampling the activations from a maze where the cheese is in the *same* location *does not* modify the behavior. Overall, these findings provide further evidence that these 11 channels affect the network's final decision mostly based on the cheese location in the maze.

We measure how frequently resampling the cheese tracking channels changes the most likely action at a decision square. If these channels mostly affect the network's behavior based on the cheese location, resampling these channels from mazes where the cheese is in the same location should only rarely affect the behavior at a decision square. Moreover, resampling from mazes where the cheese is placed in a different location would be more likely to affect the decision square behavior.

Across 200 mazes, resampling the cheese-tracking channels from mazes with a different cheese location changes the most probable action at a decision square in 40% of cases, which is much more than when

---

[3]Specifically, we compute the network activations for a different maze (maze B) where the cheese is placed in the same location as in the original maze (maze A). To "resample the activations", we replace the relevant network activations when computing the policy for maze A with the activation values computed using a network forward pass on maze B.

resampling from mazes with the same cheese location (11%). However, because resampling from mazes with the same cheese location can sometimes affect the network behavior, this suggests the cheese tracking channels also (weakly) affect the network behavior through factors other than the location of the cheese. appendix C.1 provides more evidence that these 11 channels primarily affect behavior by tracking the cheese.

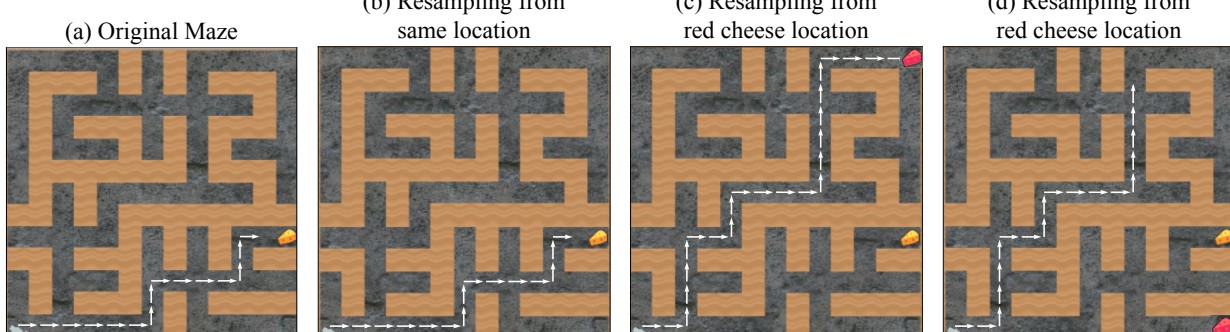

Figure 4: **Resampling cheese-tracking activations from different mazes.** (a) Unmodified network behavior. (b) Resampling these activations from other mazes with the same cheese location does not affect the policy's behavior. (c) In contrast, if we replace the activations from a maze where the cheese is placed at a different location, the network behaves as if the cheese were at that location. If the cheese-tracking channel activations are resampled from a maze where the cheese is close to the historical cheese location, the policy navigates to the cheese. (d) If the cheese-tracking channel activations are resampled from a maze where the cheese is far from the historical cheese location, the policy ignores the cheese. Please see appendix E.2 for more examples.

## 3    Controlling the Maze-Solving Policy Network

In the previous section, we showed the maze-solving policy pursues multiple, context-dependent goals. Moreover, about halfway through the network, multiple residual channels track the location of the goal. We now corroborate these findings by leveraging this understanding to design interventions that control the network's behavior. Our approach does not require collecting additional data or retraining the network, but instead utilizes existing circuits. We consider two classes of interventions: (i) manually modifying the activations in the cheese-tracking channels (§3.1); and (ii) combining activations corresponding to different forward passes (§3.2).

### 3.1    Controlling the Policy by Modifying the Cheese Channels

Previously, we identified eleven residual channels whose activations track the location of the cheese in the maze. If these activations determine network behavior by tracking the cheese location, intuitively, by modifying the activations in those channels, one should be able to modify the behavior of the policy. We now show that this is indeed the case.

First, we consider a simple, hand-designed intervention where we directly modify the activations of one of the cheese-tracking channels. Specifically, we set *just one* activation in channel 55 to a large positive value $(+5.5^4$; c.f. fig. 1c). We then consider the modified policy whose action probabilities are computed by completing the network's forward pass with this modification.

In fig. 5, we show this simple intervention retargets the policy. The network often navigates towards the region of the maze corresponding to the activation edit. We emphasize that changing just *one* activation (out of 32,768) drastically affects the behavior of the network. However, it can only partially retarget the

---

[4]We considered a range of effect sizes, and manually optimized them on the maze at seed 0.

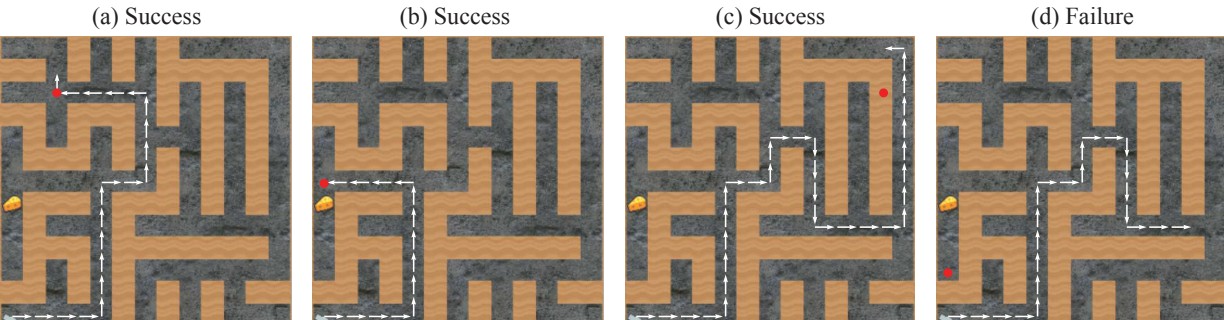

(a) Success  (b) Success  (c) Success  (d) Failure

Figure 5: **Controlling the maze-solving policy by modifying a single activation.** By modifying just a single network activation, we control where the policy navigates. We set a single activation in channel 55, one of the cheese-tracking channels, to a large positive value (+5.5; see also fig. 1c). The red dots show the location corresponding to the activation intervention, computed by linearly mapping the $16 \times 16$ activation grid to the $25 \times 25$ game grid. (a-c) Successful policy retargeting. This intervention successfully makes the policy navigate to the red dot (the targeted location) and ignore the cheese in the maze. (d) We cannot make the policy navigate to arbitrary maze-locations. See appendix E.3 for more examples.

policy. Moreover, just as the trained network sometimes ignores the cheese, we find that the retargeted network sometimes ignores the edited activation location.

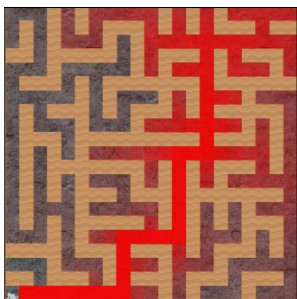

Figure 6: **Normalized path probability heatmap.** The colour of each maze square shows the *normalized path probability* for the path from the starting position in the maze to the square for the unmodified policy.

**Retargetability heatmaps.** To quantify the impact of our retargeting procedure, we compute the *normalized path probability* for paths from the starting position in a maze to each square of that maze. This is the joint probability that the policy navigates *directly* to a given square in the maze, normalized by the path distance. Specifically, we compute the geometric mean of the action probabilities leading to a given square from the start position (see eq. (2) in appendix D). In particular, for a path of $n$ steps with constant per-step action probability, the normalized path probability is independent of $n$.

We visualise *normalized path probability heatmaps* for the paths from the initial position in the maze to each square. For example, fig. 6 reveals that the policy tends to navigate towards the historical cheese location. The normalized path probabilities are higher at maze squares closer to the path between the bottom-left and the top-right corners of the maze.

**Some locations are more easily steered to.** Figure 7c shows the effect of intervening on channel 55 to target each square of the maze. That is, for each square, we retarget the policy to that square with an activation edit. We then compute the normalized path probability for the path to the target square, given the modified forward pass. For these experiments, to reduce variance, we removed cheese from the maze. In appendix D, we plot how retargetability decreases as the target location becomes increasingly far from the path to the top-right corner.

**Intervening on all 11 channels slightly improves retargetability.** Similar to the single-channel intervention, we set one of the activations of each channel to a positive value $(+1.0^5)$. Comparing the heatmaps for this intervention (fig. 7a, b) with the single-channel intervention, this edit slightly increases the normalized path-probabilities. On $13 \times 13$ mazes,[6] the averaged path probability over all legal maze squares is 0.647 from just modifying channel 55, while modifying all hypothesized cheese-tracking channels boosts the probability to 0.695.

**There are more cheese-tracking circuits.** We now compute the normalized path probabilities when targeting each square of the maze by placing the cheese in the location. If the only cheese-tracking circuits were related to the cheese-tracking channels we identified, then our activation edits would probably achieve the same retargetability as if the cheese were placed in that location. However, by actually moving the cheese around the maze, we achieve even stronger retargetability than do our activation edits (fig. 7d). This suggests that there are additional unidentified cheese-seeking mechanisms, beyond the 11 channels.

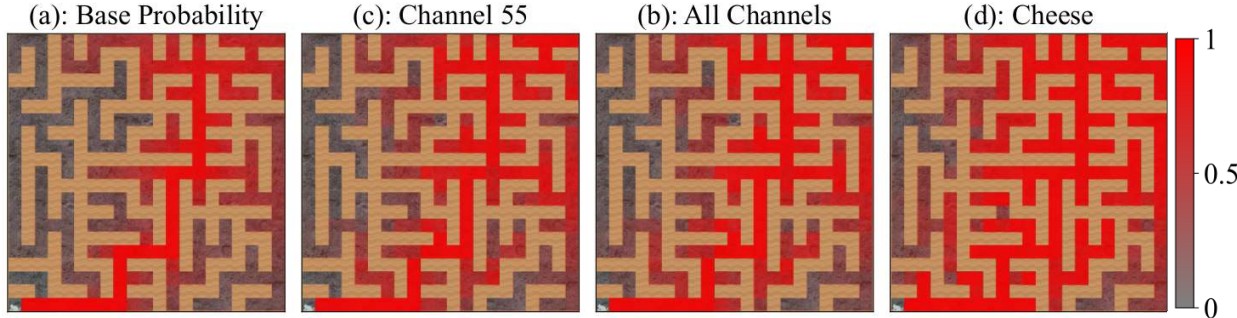

Figure 7: **Retargetability heatmaps.** The color of each maze square shows the *normalized path probability* for the path from the starting position in the maze to the square for the *modified* policy which targets that square. We modify the activations of the relevant channels so that they contain a positive value near the relevant square. The heatmap in (a) shows the base probability that each tile can be retargeted to. Intervening on a single channel (b) increases retargetability less than intervening on all cheese-tracking channels (c). However, all retargeting methods we investigated were less effective than directly moving the cheese to a tile (d), indicating that we did not find all relevant cheese-tracking circuits in the network.

## 3.2 Controlling the Policy By Combining Forward Passes

Beyond simple manual edits, we can modify the behavior of the policy by combining the activations of different forward passes of the network. These interventions do not require retraining the policy but instead leverage existing circuits. Specifically, we design different goal-modifying "steering vectors" (Subramani et al., 2022). By adding or subtracting these vectors to network activations, we modify the behavior of the network.

**Notation.** Let $\mathrm{Activ}(m, x_{\mathrm{cheese}}, x_{\mathrm{agent}}) \in \mathbb{R}^{128 \times 16 \times 16}$ be the activations after the first residual block of the second IMPALA block of the network (see fig. 11 in the appendix). At this point of the network, there are 128 channels, each of which corresponds to a $16 \times 16$ grid.[7] Activ is a function of the maze layout $m$, the position of the cheese $x_{\mathrm{cheese}}$, and the position of the agent $x_{\mathrm{agent}}$. $m \in \{0, 1\}^{25 \times 25}$ represents whether position in the maze is filled with a wall or not. Further, let $x_{\mathrm{agent}}^{\mathrm{start}}$ be the starting position of the agent in a maze.

**Reducing cheese-seeking behavior.** First, we design a "cheese vector" that weakens the policy's pursuit of cheese. The cheese vector is computed as the difference in activations when the cheese is present and not present in a given maze. Specifically, we calculate the cheese vector as $\mathrm{Activ}_{\mathrm{cheese}}(m, x_{\mathrm{cheese}}) :=$

---

[5]We optimized the magnitude of the edit to increase retargetability for both the single-channel and 11-channel interventions.
[6]Appendix D plots how retargetability decreases with maze size.
[7]The 11 cheese-tracking channels are also present at this layer.

$\text{Activ}(m, x_{\text{cheese}}, x_{\text{agent}}^{\text{start}}) - \text{Activ}(m, , x_{\text{agent}}^{\text{start}})$. For intervention coefficient $\alpha \in \mathbb{R}$,

$$\text{Activ}'(m, x_{\text{cheese}}, x_{\text{agent}}) :=$$
$$\text{Activ}(m, x_{\text{cheese}}, x_{\text{agent}}) + \alpha \cdot \text{Activ}_{\text{cheese}}(m, x_{\text{cheese}}), \tag{1}$$

and replace the original activations Activ with the modified activations Activ$'$. This intervention can be considered to define a custom bias term at the relevant residual-addition block. Figure 8 shows how subtracting the cheese vector affects the policy in a single maze.

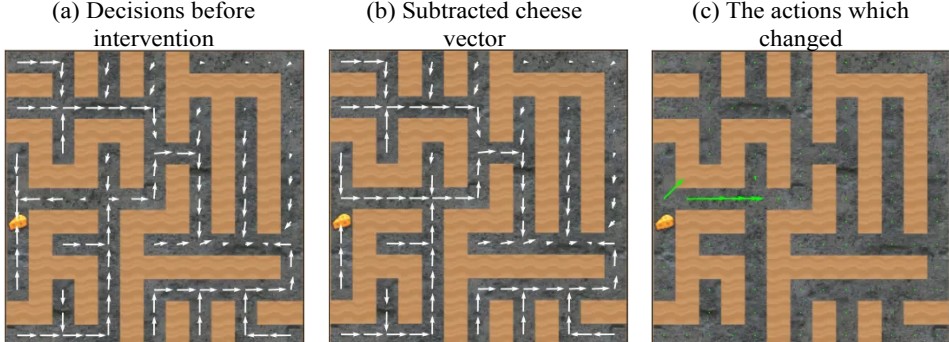

Figure 8: **Subtracting the cheese vector often appears to make the policy ignore the cheese.** We run a forward pass at each valid maze square $s$ to get action probabilities $\pi(a \mid s)$. For each square $s$, we plot a "net probability vector" with components $x := \pi(\texttt{right} \mid s) - \pi(\texttt{left} \mid s)$ and $y := \pi(\texttt{up} \mid s) - \pi(\texttt{down} \mid s)$. The policy always starts in the bottom left corner. By default, the policy goes to the cheese when near the cheese, and otherwise goes along a path towards the top-right (although it stops short of the top right corner).

**The quantitative effect of subtracting the cheese vector.** We consider 100 mazes and analyse how this subtraction affects the behavior of the policy on decision squares. Recall that decision squares are the spots of the maze where the policy must choose to navigate to the cheese or the top right corner.

In fig. 9a, subtracting the cheese vector (i.e., $\alpha = -1$)[8] substantially reduces the probability of cheese-seeking actions. Appendix C.2 shows that subtracting the cheese vector is often equivalent to the network from perceiving cheese at a given maze location, and that the cheese vector from one maze can transfer to another maze. However, *adding* the cheese vector (i.e., $\alpha = +1$) does not affect cheese-seeking action probabilities.

**Steering the policy towards the top-right corner.** We design a "top-right corner" motivational vector whose addition increases the probability that the policy navigates towards the top-right corner. We compute $\text{Activ}_{\text{top-right}}(m, x_{\text{cheese}}) := \text{Activ}(m, x_{\text{cheese}}, x_{\text{agent}}^{\text{start}}) - \text{Activ}(m', , x_{\text{agent}}^{\text{start}})$, where $m'$ is the original maze now modified so that the reachable top-right point is higher up (see fig. 28 in the appendix). Figure 10 visualizes the effect of adding the top-right vector.

In fig. 9b, we analyse the effect of different activation engineering approaches that use $\text{Activ}_{\text{top-right}}$. We find that adding $\text{Activ}_{\text{top-right}}$ (i.e., $\alpha = +1$) increases the probability the policy navigates to the top-right corner, but surprisingly, subtracting the top-right corner vector does not decrease the probability the policy navigates to the top-right. Lastly, in our experience, we can *simultaneously add the top-right vector and subtract the cheese vector in order to achieve both effects.* We were surprised that these activation vectors did not "destructively interfere" with each other.

---

[8]For both the cheese and top-right vectors, we tried optimizing $\alpha$ but found that it didn't make an appreciable difference - straightforward addition and subtraction worked best.

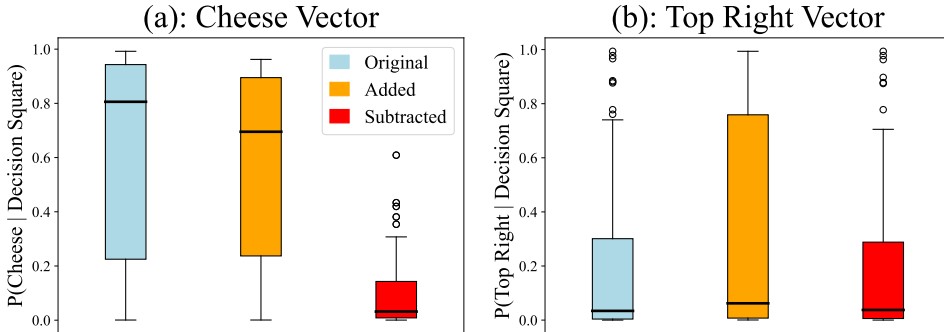

Figure 9: **Controlling the policy by combining network forward passes**. For 100 mazes, we compute the decision-square probabilities assigned to the actions which lead to the cheese (a) and to the top-right corner (b). For example, in (a), a value of 0.75 under "original" indicates that at the decision square of one maze, the unmodified policy assigns 0.75 probability to the first action which heads towards the cheese. Subtracting the cheese vector and adding the top-right vector each produce strong effects.

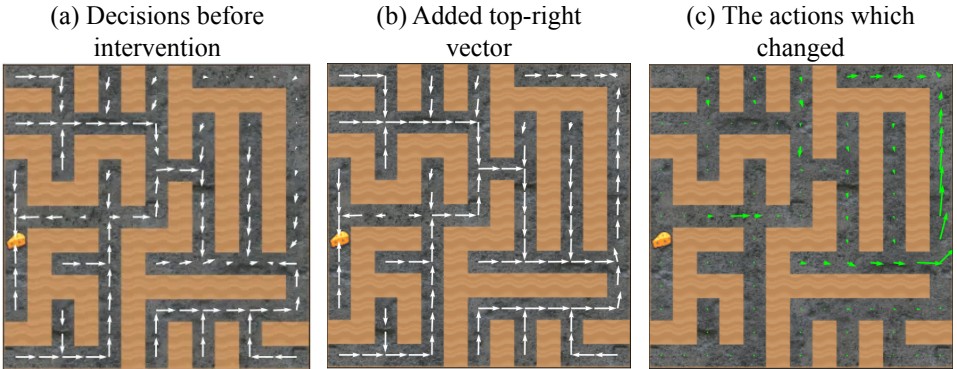

Figure 10: **Adding the "top-right vector" often appears to attract the policy to the top-right corner.** Originally, the policy does not fully navigate to the top-right corner, instead settling towards the bottom-right. After adding the top-right vector, the policy navigates to the extreme top-right.

Overall, our results demonstrate that we can control the behavior of the policy, albeit imperfectly, by combining different forward passes of the network. We were surprised, since the network was never trained to behave coherently under the addition of these "bias terms."

## 4 Related Works

**Interpretability.** Understanding AI has been a longstanding goal (e.g., Gilpin et al., 2018; Rudin et al., 2022; Zhang et al., 2021; Fan et al., 2021; Hooker et al., 2019, *inter alia*). Mechanistic approaches (Olah, 2022; Elhage et al., 2022) look to understand neural network circuits. Recently, mechanistic interpretability has helped e.g. understand grokking (Nanda et al., 2023). Lieberum et al. (2023) suggest that these approaches can scale to large models. Far less interpretability work has been done on reinforcement learning policy networks (Hilton et al., 2020; Bloom & Colognese, 2023; Rudin et al., 2022), which is our setting. To our knowledge, we are the first to interpret a non-toy policy network, and to pinpoint goal representations therein.

**Steering network behavior.** We intervened on a policy network's activations to steer its behavior, considering both hand-designed edits (§3.1) and combining forward passes (§3.2). We did not use extra training data to do so. In contrast, the most popular approaches for steering AI use training data, by *e.g.*

specifying preferences over different behaviors (Christiano et al., 2017; Leike et al., 2018; Ouyang et al., 2022; Bai et al., 2022b; Rafailov et al., 2023; Bai et al., 2022a) or through expert demonstrations (Ng et al., 2000; Torabi et al., 2018).

**Activation engineering.** Our policy interventions (§3) are examples of *activation engineering* approaches. This newly-emerging class of techniques re-use existing model capabilities. In general, these approaches can steer network behavior without behavioral data and add negligible computational overhead. For example, Subramani et al. (2022); Turner et al. (2023); Li et al. (2023) steer the behavior of language models by adding in activation vectors. In contrast, our work shows these techniques can steer a reinforcement learning policy.

**Maze navigation and goal representation in biological and artificial networks.** Spatial navigation to a goal, such as food or mates, is an ubiquitous and crucial skill for survival and is a well-studied phenomenon in neuroscience. Broadly, different types of cells play a role in navigation and path finding, such as place cells (O'Keefe, 1976), head direction cells (Taube et al., 1990a;b), grid cells (Hafting et al., 2005), and boundary cells (Lever et al., 2009). See Nyberg et al. (2022) for a thorough review of spatial goal coding in the brain. Computational models can also exhibit goal representations, such as those found through a dynamical systems lens in Singh et al. (2023).

Maze navigation is also well-studied among artificial neural networks. Some older work uses handcrafted neural networks that encode specific environmental information like distance to a goal (Bush et al., 2015), or manually replicates certain types of cells found in mammalian spatial navigation systems (Erdem & Hasselmo, 2012). More recent work does not use such manual methods. Banino et al. (2018) use deep reinforcement learning agents to solve navigation tasks, finding vector-based goal representations through certain 'grid-like' representations that develop in the network. Wijmans et al. (2023) also test for goal-like representations while giving maze-solving networks minimal information. Somewhat similarly, we find vectors which seem to activate goals in pre-trained deep reinforcement learning models. We use these vectors to effectively steer the network at inference time.

## 5 Discussion

Activation steering approaches for controlling a model are well-studied [9] for different problems, as they propose a method for customization not based on potentially expensive fine-tuning strategies (Turner et al., 2023). They have been used for reducing toxicity (Liu et al., 2024), sentiment steering for style transfer (Konen et al., 2024), and for enhanced LLM red teaming (Wang & Shu, 2023). Activation steering can also be combined with other techniques such as sparse autoencoders Bricken et al. (2023), to steer a model in a more interpretable way (Nanda et al., 2024).

We lay the foundation for using these techniques to steer policy networks, especially in cases where safe navigation and control may be essential. For example, in safety-critical systems such as autonomous vehicles, explainable reinforcement learning systems are important for not just safety, but also social acceptance of this new technology by the broader public (Atakishiyev et al., 2024). If explainability methods can isolate safety relevant concepts to internal representations, our method can then be used for efficient steering and control without re-training. Similarly, in robotics applications, instead of requiring modifications to an optimization objective or training scheme with the intent to build a safer system (García & Shafie, 2020; Pham et al., 2018), we speculate that activation steering could be used at inference to steer a model towards not only its goal (as we have tested) but also away from dangerous scenarios. More broadly, activation steering often works with a few dozen supervised datapoints (Li et al., 2023), which is promising for real-world domains where data collection is expensive.

---

[9]These activation steering techniques were published after the first revision of this work.

## 6    Conclusion

We studied the goals and goal representations of a pretrained policy network, following a model organisms paradigm. We found that this network pursues multiple, context-dependent goals (§2.1). We found 11 channels that track the location of the cheese within each maze (§2.2). By modifying just a *single* activation, or by adding in simple activation vectors, we steered which goals the policy pursued (§3). Our work shows the goals of this network are redundant, distributed, and retargetable. In general, policy networks may be well-understood as pursuing multiple context-dependent goals.

## 7    Impact Statement

This work makes progress toward understanding and controlling the internal goal representations within reinforcement learning policy networks. We studied a maze navigation policy that exhibits harmful goal misgeneralization during deployment and how to steer the policy towards its intended goal.

These techniques could potentially be used by malicious actors to misdirect autonomous systems. However, with appropriate safeguards, the ability to understand and direct policy networks could also have benefits. For example, this approach might be used for alignment verification, safer exploration in RL, or correcting unwanted bias quickly rather than retraining models. By developing theories of the goals and motivations latent in AI systems, we may be able to build more robust and beneficial systems.

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

## A  Training Details

We did not train the network which we studied. Langosco et al. (2023) trained 15 maze-solving 3.5M-parameter deep convolutional network using Proximal Policy Optimization (Schulman et al., 2017). For

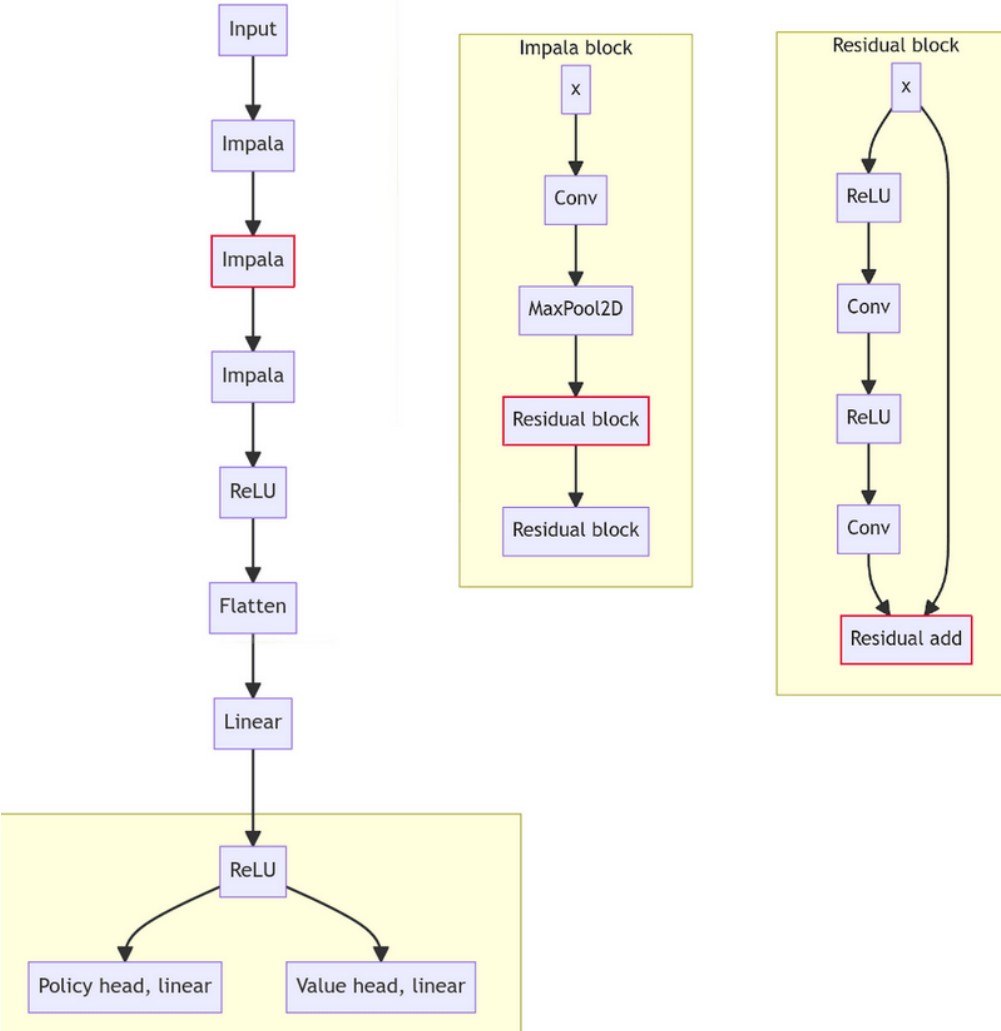

Figure 11: A high-level visualization of the policy network architecture, using IMPALA blocks from Espeholt et al. (2018). The red-outlined layer contains the 11 goal-tracking channels and is where the "cheese vector" and "top-right vector" were applied. For more details, refer to Langosco et al. (2023).

each of $n = 1, \ldots, 15$, network $n$ was trained in mazes where cheese was randomly placed in a free tile in the top-right $n \times n$ squares of the maze. We primarily study the $n = 5$ network.

When the policy reached the cheese, the episode terminated and a reward of $+10$ was recorded. Each model was trained on 100K procedurally generated levels over the course of 200M timesteps. Figure 11 diagrams the high-level architecture.

At each timestep, the policy observes a $64 \times 64$ RGB image, as shown by fig. 13. The policy has five actions available: $\mathcal{A} := \{\uparrow, \rightarrow, \downarrow, \leftarrow, a_{\text{do nothing}}\}$.

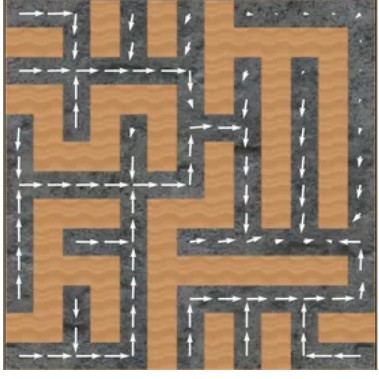 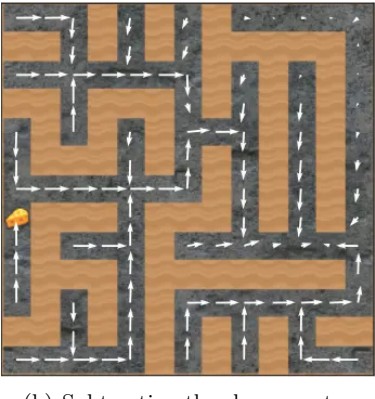 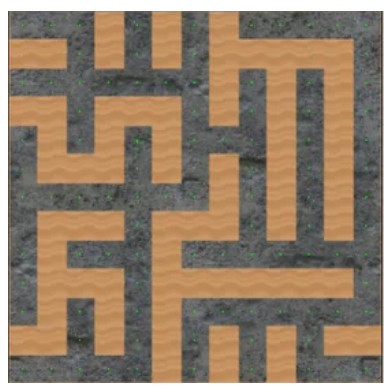

(a) No cheese present | (b) Subtracting the cheese vector | (c) The actions which changed

Figure 12: In seed 0, subtracting the cheese vector is behaviorally equivalent to hiding the cheese.

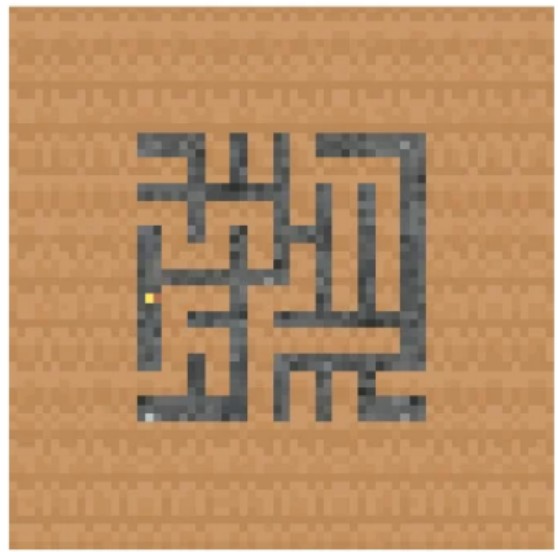 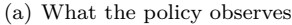 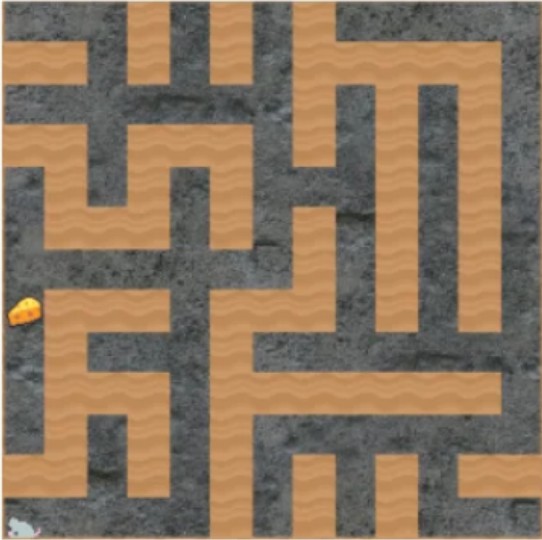

(a) What the policy observes | (b) Human-friendly visualization

Figure 13: The mazes are defined on a $25 \times 25$ game grid. For some mazes, the accessible maze is smaller, and the rest is padded. Furthermore, the network observes a $64 \times 64$ RGB image (fig. 13(a)). In contrast, we visualize mazes as in fig. 13(b): without padding, and as a higher-resolution image.

## B  Behavioral Statistics

We wanted to better understand the generalization behavior of the network. During training, the cheese was always in the top-right $n \times n$ corner. During testing, the cheese can be anywhere in the maze. In the test distribution, visual inspection of sampled trajectories suggested that the network has goals related to at least two historical reward proxies: the cheese, and the top-right corner.

To understand generalization behavior, we wanted to understand which maze features correlate with the network's decision to pursue the cheese or the corner. For each of the 15 pretrained networks, we uniformly randomly sampled (without replacement) 10,000 maze seeds between 0 and 1e6. We sampled a rollout in each seed. We recorded various statistics of the maze and rollout, such as whether the agent reached the cheese. We then discarded mazes without decision squares (fig. 14), since in these mazes the policy does not have to choose between the cheese or the corner. We also discarded mazes with cheese in the top-right $5 \times 5$

corner, because i) we wanted to test generalization behavior, and ii) the cheese is probably just a few steps from the decision square. This left us with 5,239 rollouts.

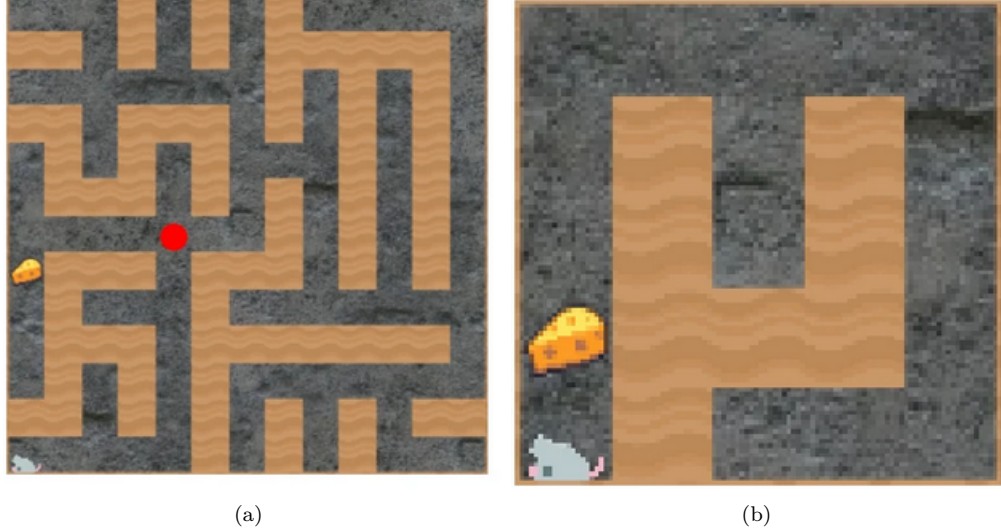

(a)                                                                 (b)

Figure 14: A *decision square* is the square where there is divergence between the paths to the cheese and to the top-right corner. In the first maze, the decision square is shown by a red dot. The second maze does not have a decision square.

We considered a range of metrics. We considered two notions of distance and five pairs of maze landmarks, and then measured their 10 possible combinations. The distances comprised:

1. The Euclidean $L_2$ distance in the game grid, $d_2$.

2. The maze path distance, $d_{\text{path}}$. Each maze is simply connected, without loops or "islands." Therefore, there is a unique shortest path between any two maze squares.

The pairs of maze landmarks were:

1. The top-right $5 \times 5$ region and the cheese.

2. The top-right $5 \times 5$ region and the decision square.

3. The cheese and the decision square.

4. The cheese and the top-right square.

5. The decision square and the top-right square.

Figure 15 visualizes four of these feature combinations.

We also regressed upon the $L_2$ norm of the cheese coordinate within the $25 \times 25$ game grid (where the bottom-left corner is the origin $(0,0)$). All else equal, larger coordinate norm is correlated with the cheese being closer to the top-right corner (fig. 16).

To discover which of the 11 features are predictive, we trained single-variable regression models using $\ell_1$-regularized logistic regression. As a baseline, always predicting that the agent gets the cheese yields an accuracy 71.4%. Among the 11 variables investigated, 6 variables outperformed this baseline (table 1). The rest performed worse than the no-regression baseline (table 2).

| Variable | Prediction accuracy |
|---|---|
| $d_2(\text{cheese}, \text{top-right } 5 \times 5)$ | 0.775 |
| $d_2(\text{cheese}, \text{top-right square})$ | 0.773 |
| $d_2(\text{cheese}, \text{decision square})$ | 0.761 |
| $d_{\text{path}}(\text{cheese}, \text{decision square})$ | 0.754 |
| $d_{\text{path}}(\text{cheese}, \text{top-right } 5 \times 5)$ | 0.735 |
| $d_{\text{path}}(\text{cheese}, \text{top-right square})$ | 0.732 |

Table 1: Variables that outperform the no-regression baseline of 71.4%. We found that these variables have negative regression coefficients, which matched our expectation that increased distance generally discourages cheese-seeking behavior.

| Variable | Prediction accuracy |
|---|---|
| $\|\text{cheese coord}\|_2$ | 0.713 |
| $d_2(\text{decision square}, \text{top-right square})$ | 0.712 |
| $d_{\text{path}}(\text{decision square}, \text{top-right square})$ | 0.709 |
| $d_{\text{path}}(\text{decision square}, \text{top-right } 5 \times 5)$ | 0.708 |
| $d_2(\text{decision square}, \text{top-right } 5 \times 5)$ | 0.708 |

Table 2: Variables that underperform the no-regression baseline of 71.4%.

## B.1 Handling multicollinearity

Table 1 yielded 6 individually predictive features. However, many of these features are strongly correlated (fig. 17 and fig. 18). In these situations, we must take extra care when regressing on all 6 variables and then interpreting the regression coefficients.

We measure the variance inflation factor (VIF) in order to quantify the potential multicollinearity (James et al., 2013). VIF greater than 4 is considered indicative of multicollinearity.

| Features | VIF |
|---|---|
| $d_2(\text{cheese}, \text{decision square})$ | 5.16 |
| $d_2(\text{cheese}, \text{top-right})$ | 107.96 |
| $d_2(\text{cheese}, 5 \times 5 \text{ top-right})$ | 107.52 |
| $d_{\text{path}}(\text{cheese}, \text{decision square})$ | 5.43 |
| $d_{\text{path}}(\text{cheese}, 5 \times 5 \text{ top-right})$ | 8.01 |
| $d_{\text{path}}(\text{cheese}, \text{top-right})$ | 7.88 |

Table 3: Variation inflation factors for the 6 predictive variables. These variables display large multicollinearity.

## B.2 Assessing stability of regression coefficients

With the multicollinearity in mind, we perform an $\ell_1$-regularized multiple logistic regression on the 6 predictive variables to assess their stability and importance. We compute results for 2,000 randomized test/train splits. The results are shown in table 4.

Over the 2,000 regressions, the three italicized variables in table 4 are the only variables to not sign flip. To further validate these results, we found that our conclusions held on another dataset of 10K randomly seeded mazes.

We also regressed on 200 random subsets of the 6 variables. The aforementioned 3 variables never experienced a sign flip, strengthening our confidence that multicollinearity has not distorted our original regressions. Taken together, this is why section 2.1 presents results for these three features.

| Attribute | Coefficient |
|---|---|
| Steps between cheese and top-right $5 \times 5$ | $-0.003$ |
| Euclidean distance between cheese and top-right $5 \times 5$ | $0.282$ |
| Steps between cheese and top-right square | $1.142$ |
| *Euclidean distance between cheese and top-right square* | *$-2.522$* |
| *Steps between cheese and decision-square* | *$-1.200$* |
| *Euclidean distance between cheese and decision-square* | *$0.523$* |
| Intercept | $1.418$ |

Table 4: Coefficients from the initial $\ell_1$-regularized multiple regression. The 3 variables from section 2.1 are italicized. Regression accuracy is 84.1%.

### B.2.1   Regressing on the stable features

A regression using only the three stable variables retains an accuracy of 82.4%, averaged over 10 splits (table 5). This is a 1.7% accuracy drop from the initial multiple regression on all 6 variables (table 4).

| Attribute | Coefficient |
|---|---|
| Euclidean distance between cheese and top-right square | $-1.405$ |
| Steps between cheese and decision-square | $-0.577$ |
| Euclidean distance between cheese and decision-square | $-0.516$ |
| Intercept | $1.355$ |

Table 5: Regression accuracy is 82.4%. Coefficients when regressing only on stable variables. We caution that our analysis is not meant to hinge on the *coefficient magnitudes*, which are often contingent and unreliable metrics. Instead, we think their *sign and stability* are better correlational evidence for the impact of these features on the policy's decisions.

We found that while adding a fourth variable (from the 6 above) can increase regression accuracy slightly, the fourth variable has flipped sign. We interpret this as further evidence that the other variables do not represent interpretable, meaningful influences on the policy's decision-making.

### B.3   Speculation on causality

Figure 19 demonstrates the large impact of increasing path distance to cheese, while holding constant the other two stable variables.

Table 6 examines how dropping each stable variable impacts the regression accuracy. This provides evidence on the predictive importance of each feature.

Considering both the qualitative and statistical findings, we have strong confidence that $d_{\text{step}}$(cheese, decision-square) influences decision-making. We are more cautious about $d_2$(cheese, decision-square), although its removal causes a notable accuracy drop similar to that of $d_{\text{step}}$(cheese, decision-square), a variable we are confident about. Overall, we suspect *both* of these variables affect decision-making, even though optimal policies would generally only depend on step distance (due to the discounting term).

| Regression variables | Accuracy |
|---|---|
| $d_2$(cheese, top-right square) $d_{\text{step}}$(cheese, decision-square) $d_2$(cheese, decision-square) | 82.4% |
| $d_{\text{step}}$(cheese, decision-square) $d_2$(cheese, decision-square) | 75.9% |
| $d_2$(cheese, top-right square) $d_2$(cheese, decision-square) | 81.9% |
| $d_2$(cheese, top-right square) $d_{\text{step}}$(cheese, decision-square) | 81.7% |

Table 6: Regression accuracy after dropping variables. Similar drops in accuracy occur for dropping $d_{\text{step}}$(cheese, decision-square) and $d_2$(cheese, decision-square).

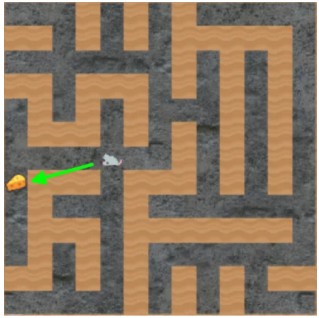

(a) $L_2(\text{decision sq.}, \text{cheese})$

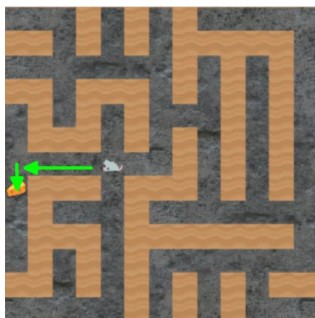

(b) Steps from the decision square to the cheese

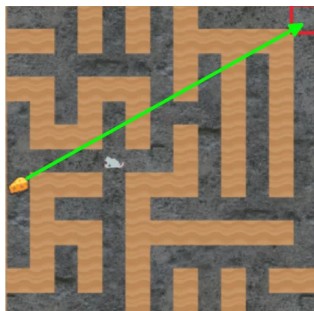

(c) $L_2(\text{cheese}, \text{top-right sq.})$

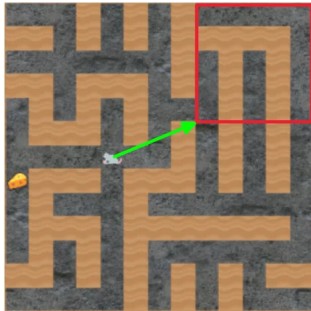

(d) $L_2(\text{decision sq.}, \text{top-right} 5x5$

Figure 15: Four of the features we regress upon.

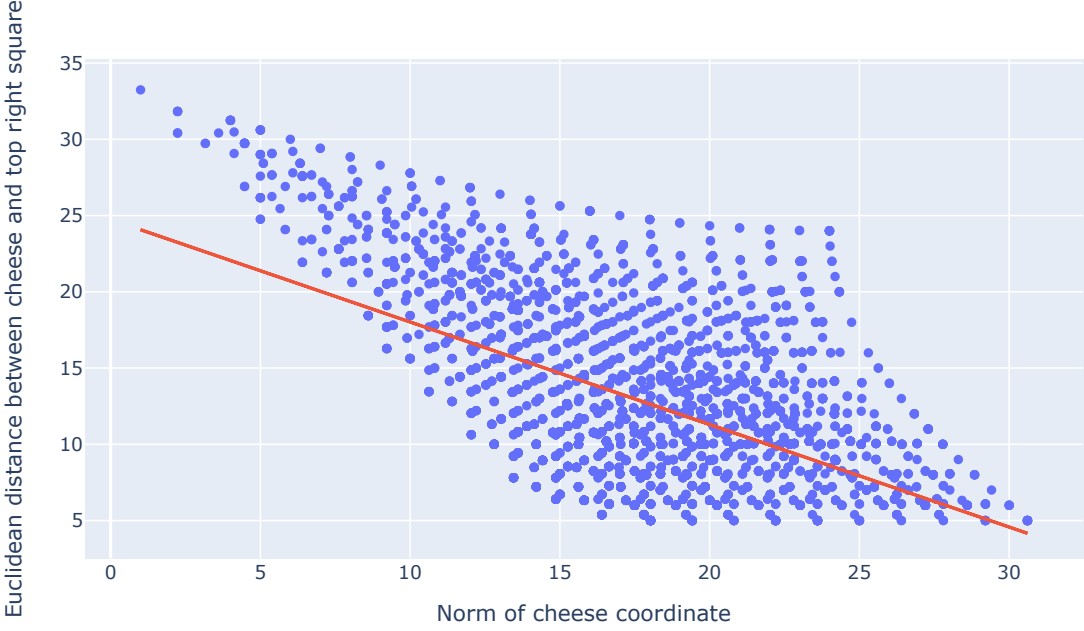

Figure 16: Among mazes with decision squares, there is a Pearson correlation of $-.550$ between the norm of the cheese coordinate and the distance. That is, the larger the norm, the closer the cheese is (in $L_2$) to the top-right square of the $25 \times 25$ grid.

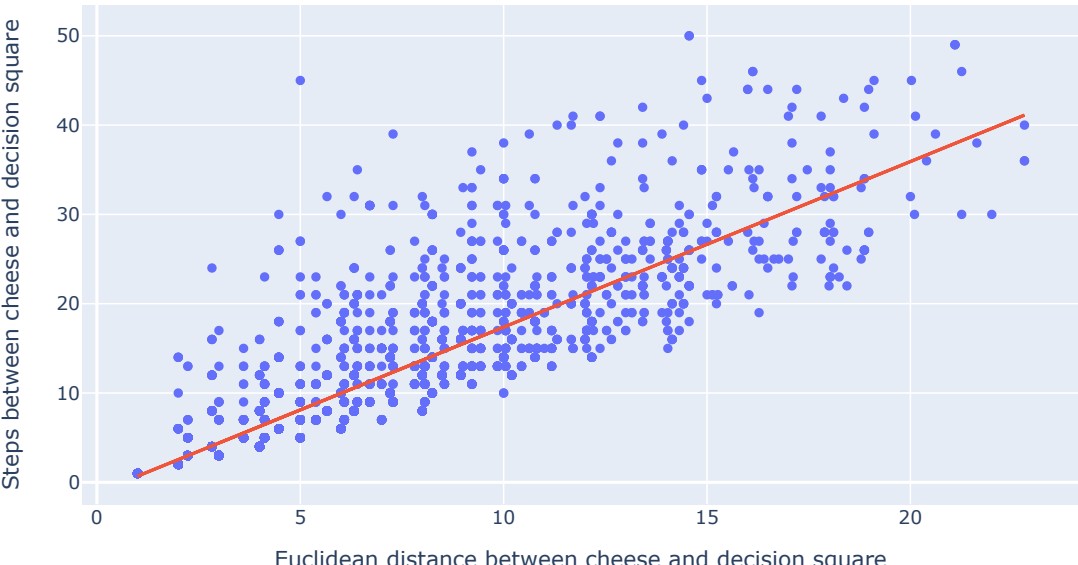

Figure 17: Among mazes with decision squares, there is a Pearson correlation of .886 between these two distances.

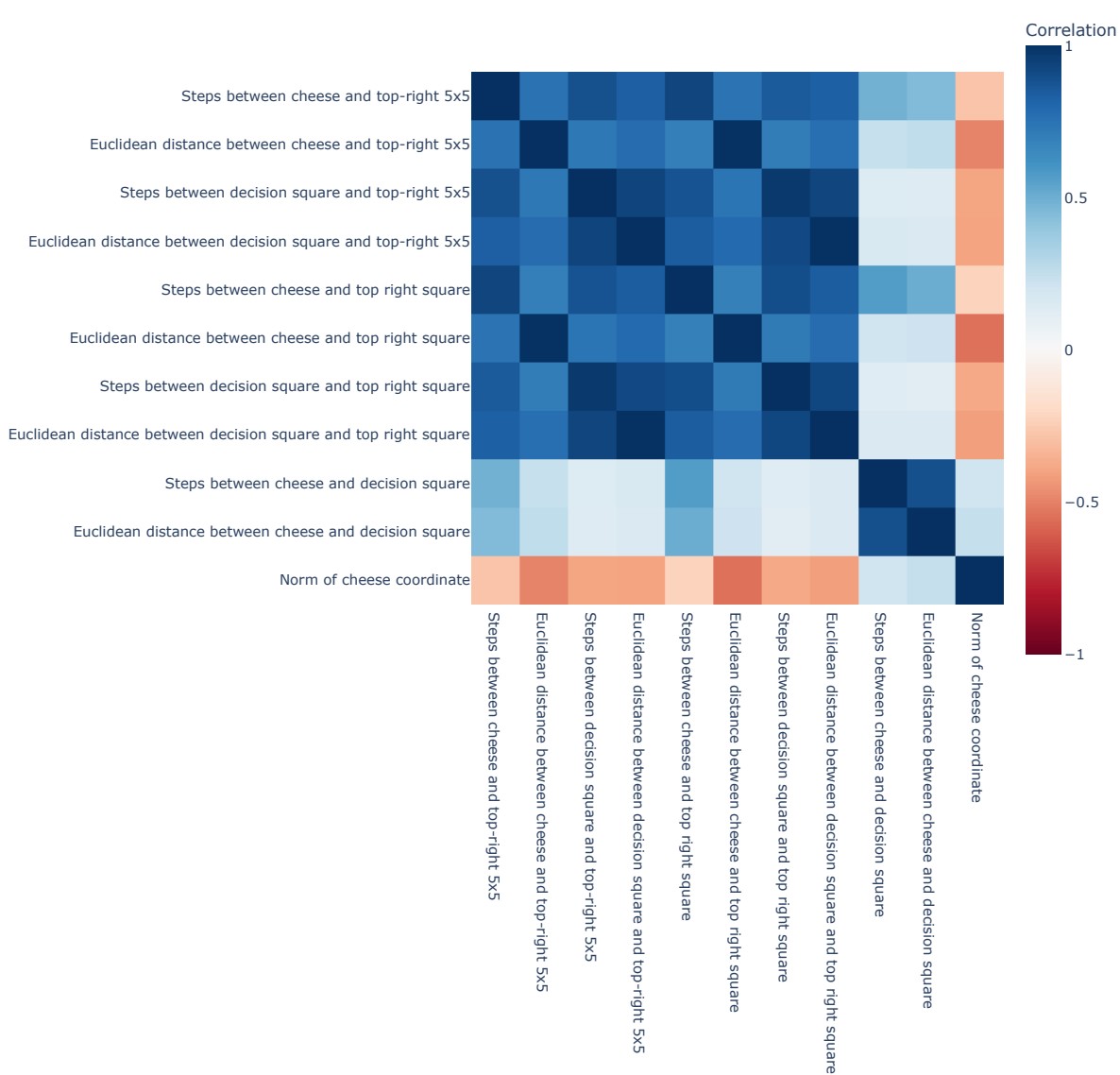

Figure 18: The correlations between maze features, considering only mazes with decision squares.

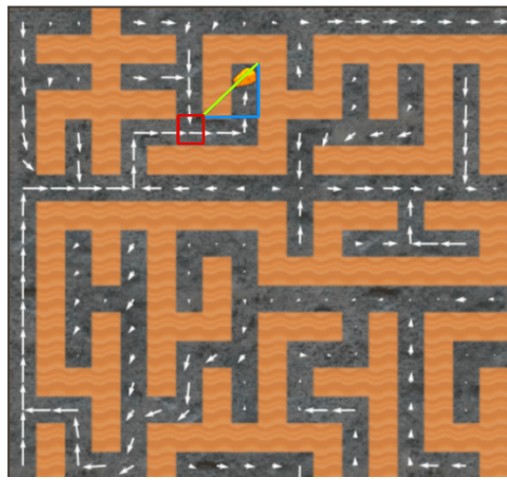

(a) Low $d_{\mathrm{path}}(\text{decision sq.}, \text{cheese})$

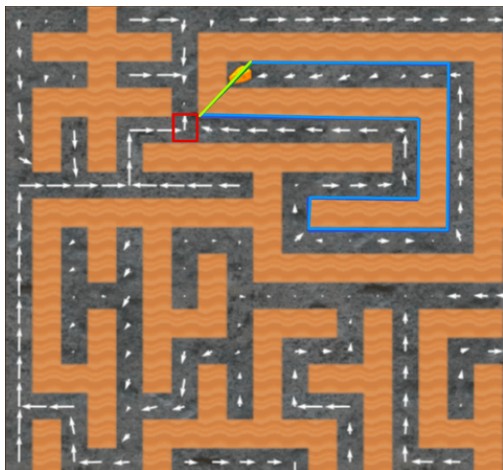

(b) High $d_{\mathrm{path}}(\text{decision sq.}, \text{cheese})$

Figure 19: **The causal effect of increased path distance to cheese.** We illustrate policy behavior using the "vector field" view introduced by fig. 8. The decision square is boxed in red. Holding constant $d_2(\text{decision sq.}, \text{cheese})$ (in green) and $d_2(\text{decision sq.}, \text{top-right sq.})$, an increase in $d_{\mathrm{path}}(\text{decision sq.}, \text{cheese})$ (in blue) makes the policy far less likely to pursue the cheese.

### B.4 Data from other models

We briefly examined other pretrained models from Langosco et al. (2023). For each of the models trained on cheese in the top-right $n \times n$ corner for $n = 3, \ldots, 15$, we run the $\ell_1$-regularized logistic regression on the three stable variables. Each setting regresses upon about 550 mazes.

| Size | Steps from decision sq. to cheese | $L_2(\text{decision sq.}, \text{cheese})$ | $L_2(\text{top-right sq.}, \text{cheese})$ |
|---|---|---|---|
| 3 | $-0.681$ | $0.000$ | $-1.935$ |
| 4 | $-0.276$ | $-0.476$ | $-1.438$ |
| **5** | $\mathbf{-0.348}$ | $\mathbf{-0.745}$ | $\mathbf{-1.278}$ |
| 6 | $-1.606$ | $-0.324$ | $-1.361$ |
| 7 | $-1.087$ | $-0.208$ | $-1.670$ |
| 8 | $-0.759$ | $-0.606$ | $-1.833$ |
| 9 | $-0.933$ | $-0.112$ | $-1.943$ |
| 10 | $-1.051$ | $-0.040$ | $-2.075$ |
| 11 | $-1.102$ | $0.000$ | $-1.212$ |
| 12 | $-0.860$ | $0.000$ | $-1.732$ |
| 13 | $-1.002$ | $-0.045$ | $-2.286$ |
| 14 | $-0.743$ | $0.150$ | $-1.394$ |
| 15 | $-0.663$ | $-0.402$ | $-1.726$ |

Table 7: **Regression coefficient signs are somewhat stable across $n$ settings.** The regression coefficients found by $\ell_1$-regularized logistic regression. A *size* of $n$ indicates that the cheese was spawned in the top-right $n \times n$ region of each maze during training. Each size value corresponds to a separate pretrained model from Langosco et al. (2023). Recall that this work mostly examines the $n = 5$ case (and thus that row is bolded). The $n = 1, 2$ cases did not pursue cheese outside of the top-right $5 \times 5$ region, and so are omitted.

## C  Additional Experiments

### C.1  Causal scrubbing

In fig. 4, we explored the results of resampling channel activations from other mazes. In this subsection, we motivate this technique and explore additional quantitative results.

Chan et al. (2022) introduce *causal scrubbing*. The basic idea is: If the important computation performed by part of a network only depends on a few input features (like the presence of cheese at a certain coordinate), then randomizing other input features shouldn't degrade performance.

We test the hypothesis that, at the forward pass location highlighted by fig. 11, the residual channels $7, 8, 42, 44, 55, 77, 82, 88, 89, 99, 113$ are some function $f$ of the absolute position of cheese in the input image (fig. 20). We call these the "cheese-tracking" channels. If this hypothesis is true, then we should be able to replace the cheese-tracking activations with the activations from a another maze with cheese in the same absolute location, without disrupting behavior. We will call this the *same-cheese* condition. Alternatively, we could resample activations from any other maze (not requiring the cheese to be in the same location). This is the *random-cheese* condition.

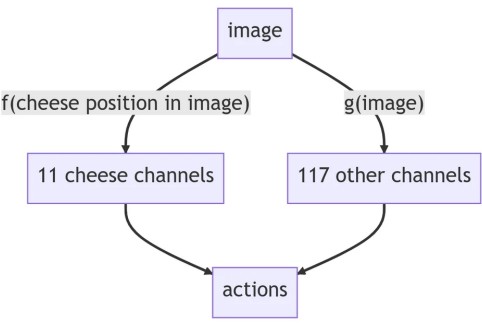

Figure 20: The computational graph which we test.

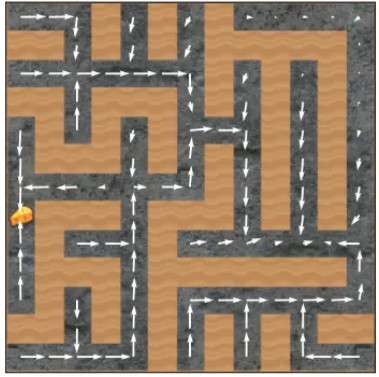 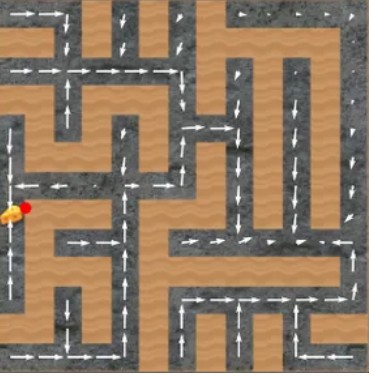 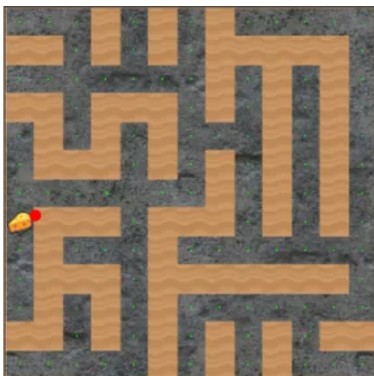

(a) The original actions      (b) Fixed-cheese resampling      (c) The actions which changed

Figure 21: We resample activations for the 11 channels which we hypothesized to track cheese: Using the vector field visualization introduced by fig. 8, behavior is almost invariant to resampling activations from another maze with cheese in the same location (shown as a red dot). This invariance is demonstrated by the imperceptible green difference arrows in fig. 21(c).

If the same-cheese condition changes the action probabil-
ities less than the random condition, this is evidence for
our hypothesis (shown in fig. 20). To quantify change in action probabilities, we perform the following procedure for each of the first 30 maze seeds:

1. Compute the action probabilities at every free square in the maze. These are the *base* probabilities.

2. For each channel:

   (a) Generate another maze with cheese in the same location, and a totally random maze seed.
   (b) Record the activations for each.

3. Substituting the appropriate channel activations during the forward pass, compute the same-cheese and random-cheese action probabilities for each free square in the maze.

4. Compute the average absolute difference[10] between action probabilities between:

   (a) The fixed-cheese and base probabilities, and
   (b) The random-cheese and base probabilities.

Figure 21 and fig. 22 show the impact of the two resampling conditions.

As a control, we further compare to the effects of resampling activations to a random subset of 11 channels (excluding those we are already testing). Table 8 shows the results.

|  | Same cheese location | Random cheese location |
|---|---|---|
| 11 "cheese-tracking" channels | 0.88% | 1.26% |
| 11 randomly selected channels[11] | 0.60% | 0.54% |

Table 8: Average change in action probabilities given different resampling procedures. The average is taken across the first 30 maze seeds.

table 8's quantitative results seem somewhat weaker than expected if fig. 20's hypothesis were entirely accurate. However, our channel selection could inherently be biased towards those that have a more significant

---

[10]I.e. the total-variation distance.

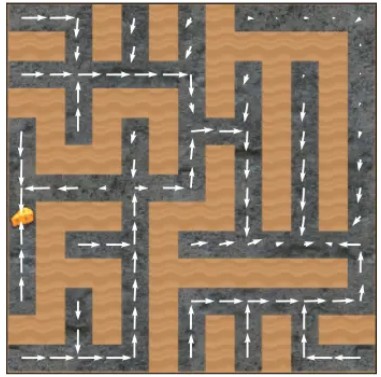 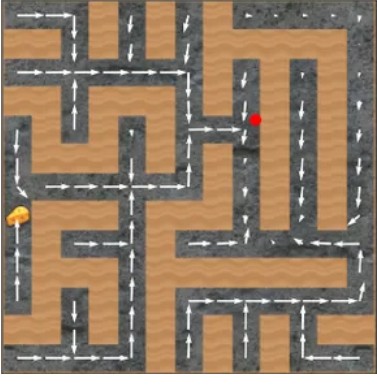 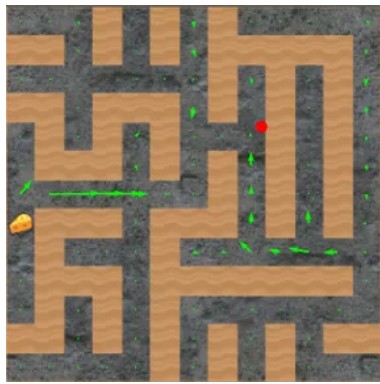

(a) The original actions      (b) Random-cheese resampling      (c) The actions which changed

Figure 22: Behavior significantly changes when resampling "cheese-tracking" activations from another maze with cheese in a different location (shown as a red dot).

impact on action probabilities. We found some additional evidence (not included in this manuscript) supporting this hypothesis. Furthermore, the total variation distance statistic does not account for the *distribution* of changes in action probabilities—whether the changes are distributed across multiple minor adjustments or concentrated in a few pivotal locations.

## C.2 Subtracting the cheese vector probably removes the ability to see cheese at a location

### C.2.1 Subtracting the cheese vector often has similar effects to hiding the cheese

Figure 23 and fig. 24 demonstrate our experience that "subtracting the cheese vector" is often behaviorally equivalent to "hide the cheese from view." If true, this allows us to interpret the effect of subtracting the steering vector. This high-level understanding could lead to further insights into the learned computational structure of the policy network which we studied.

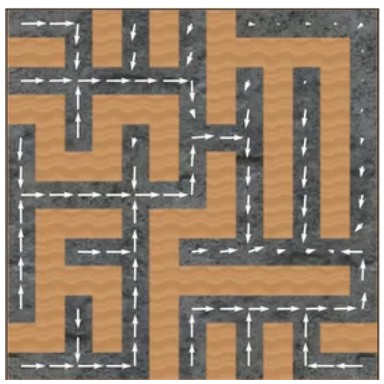 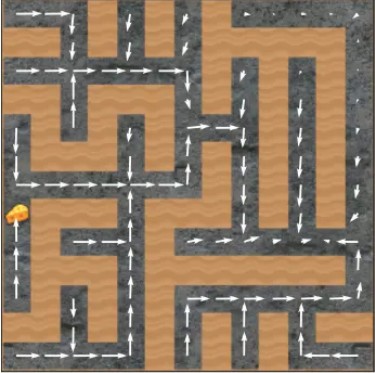 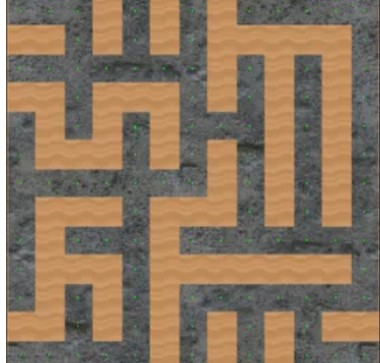

(a) No cheese present      (b) Subtracting the cheese vector      (c) The actions which changed

Figure 23: In seed 0, subtracting the cheese vector is behaviorally equivalent to hiding the cheese.

However, in a few mazes (as in fig. 25), the cheese vector is not functionally equivalent to hiding the cheese. This suggests that "hides the cheese location" is an important approximation to the function of the cheese vector, but is not the whole story.

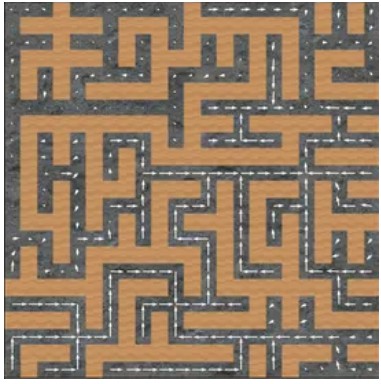 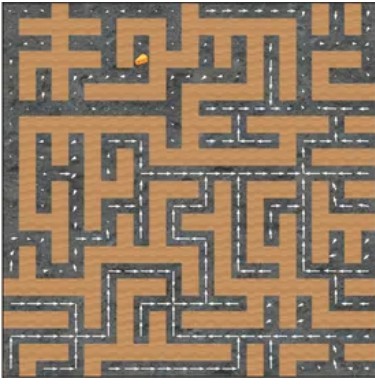 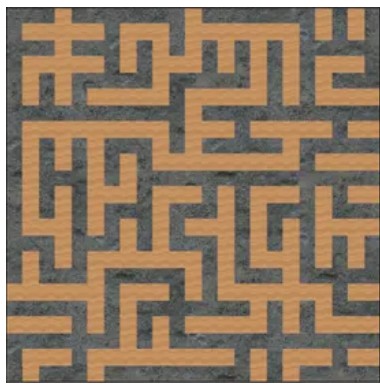

(a) No cheese present
(b) Subtracting the cheese vector
(c) The actions which changed

Figure 24: In seed 12, subtracting the cheese vector is behaviorally equivalent to hiding the cheese.

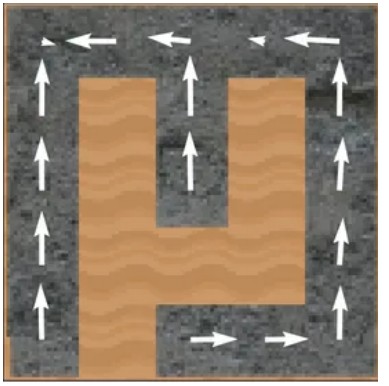 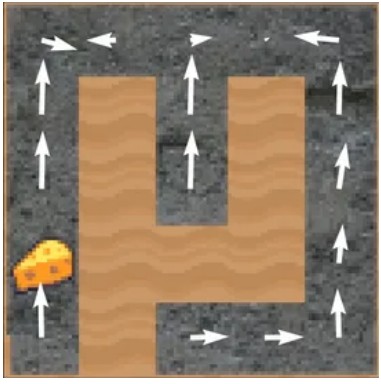 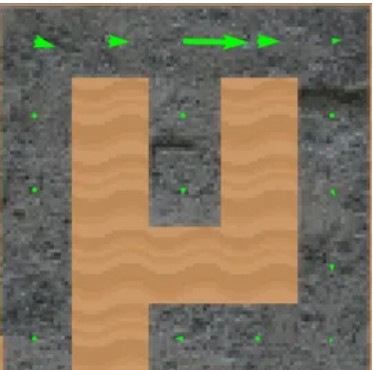

(a) No cheese present
(b) Subtracting the cheese vector
(c) The actions which changed

Figure 25: In seed 7, subtracting the cheese vector is *not* behaviorally equivalent to hiding the cheese.

### C.2.2 Cheese vectors transfer to mazes with similarly-placed cheese

Suppose we compute a cheese vector for maze A. Can we also subtract the vector during navigation of some other maze B? Our qualitative results indicate "yes, but only if the cheese is within about 2 tiles of its original position."

Figure 27 shows that the cheese vector computed on seed 0 also works on seed 795 (which has cheese at the same location; fig. 26). This suggests that the cheese vector is a function of cheese location, and not of e.g. the placement of walls in the maze.

### C.3 Computation of steering vectors

We discuss the "contrast pair" (Burns et al., 2022) we used to compute the top-right steering vector (as defined in section 3.2).

Empirically, having a path to the extreme top-right increases the policy's attraction towards the top-right corner. We hypothesize that the policy tracks the "priority" of navigating to the top right corner, and adding in the top-right vector increases that priority.

Seed 0

Seed 795

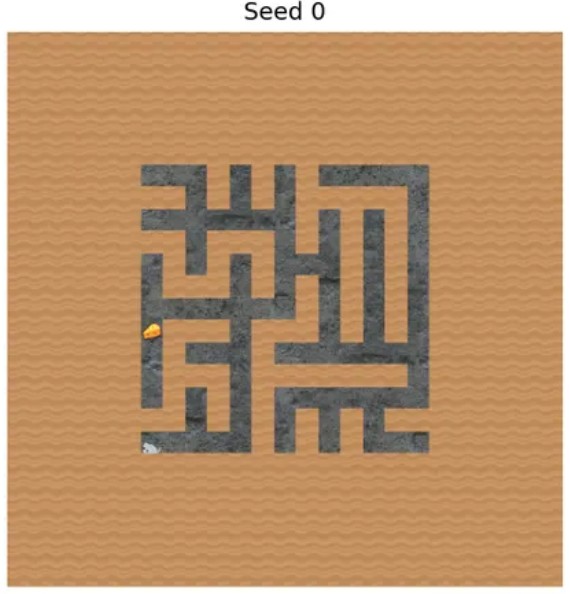
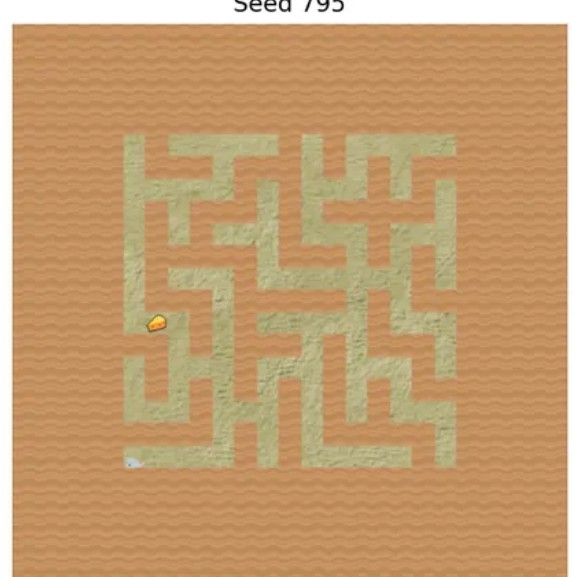

Figure 26: Two seeds with cheese at the same position.

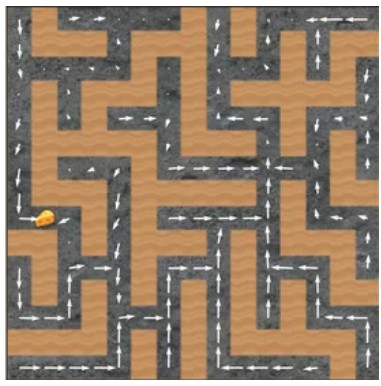
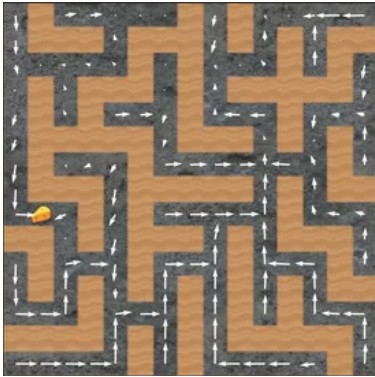
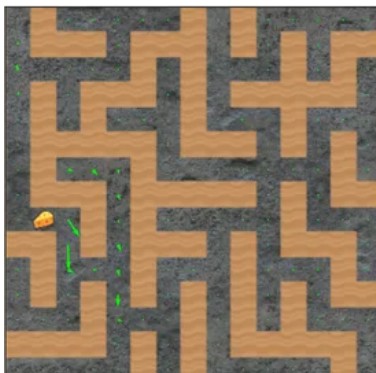

(a) No cheese present

(b) Subtracting cheese vector from seed 0

(c) The actions which changed

Figure 27: In seed 795, subtracting the cheese vector from seed 0 still makes the policy ignore the cheese.

## D    Quantitative Analysis of Retargetability

The *top-right path* is the path from the policy's starting location in the bottom left, to the top right corner. Figure 29 shows a heatmap of each square's path distance from the top-right path.

We find that the probability of successfully retargeting the policy decreases as the path distance from the top-right path increases, as in fig. 30. These results corroborate the data and heatmaps discussed in §3.1.

We analyze the policy's retargetability on the first 100 maze seeds. Suppose we wish to compute retargetability to state $s_t$ in the maze. We do this as follows: Given initial state $s_0$ and target state $s_t$ (neither containing a wall), the *normalized path probability* is

$$P_{\text{path}}(s_t \mid \pi) := \sqrt[t]{\prod_{i=0}^{t-1} \pi(a_i \mid s_i)}, \tag{2}$$

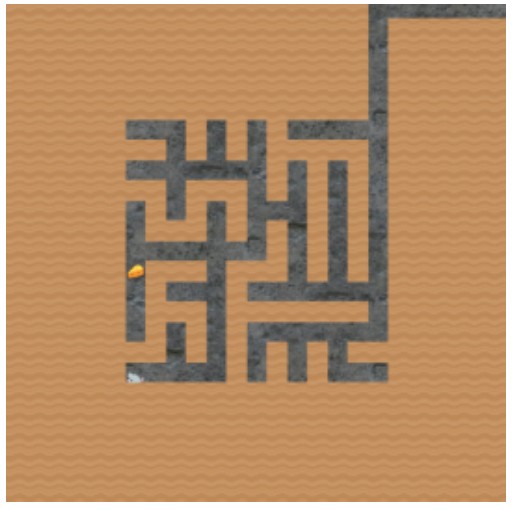 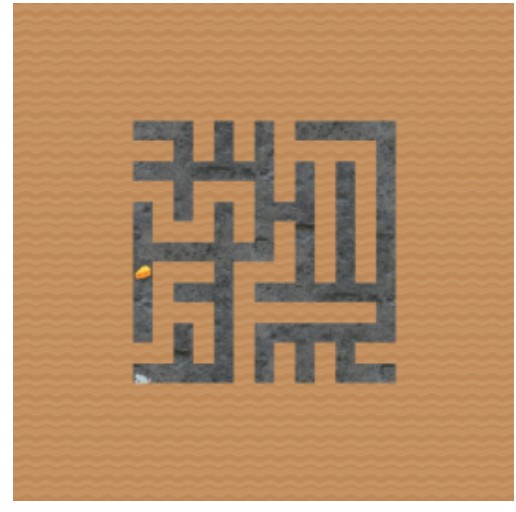

(a) The maze modified to have a reachable top-right-most square.

(b) The original maze.

Figure 28: We run a forward pass on both mazes. The top-right vector consists of the activations for (a) minus the activations for (b), at the relevant layer (see fig. 11). This operation can be performed algorithmically for any maze, although we found that the top-right vector from maze A often transfers to other mazes.

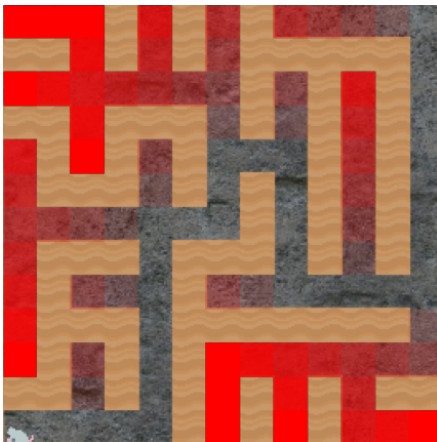

Figure 29: Each square's path distance from the top-right path (which is not at all reddened). The brighter the red, the greater the distance.

where $s_0, s_1, \ldots, s_t$ is the unique[12] shortest path between $s_0$ and $s_t$, navigated by actions $a_i$. If $s_0 = s_t$, then $P_{\text{path}}(s_t \mid \pi) := \pi(\texttt{no-op} \mid s_0)$.

For each of the 100 maze seeds, we remove the cheese from the maze. Then fig. 7 computes the following statistics for each target square $s_t$:

**Base Probability**  Computes eq. (2) for the unmodified policy $\pi$.

**Channel 55**  $\pi$ is modified to incorporate an $\alpha = +5.5$-strength intervention at the channel-55 activation corresponding to the location of $s_t$.

**Effective Channels (not shown in fig. 7)**  An $\alpha = +2.3$ intervention on channels $\{8, 55, 77, 82, 88, 89, 113\}$.

---

[12]Because the maze is simply connected.

**All Channels** An $\alpha = +1.0$ intervention on channels $\{7, 8, 42, 44, 55, 77, 82, 88, 89, 99, 113\}$.

**Cheese** The unmodified policy $\pi$ is retained, but cheese is placed at $s_t$, and eq. (2) is computed according to the new state observations $s_i$.

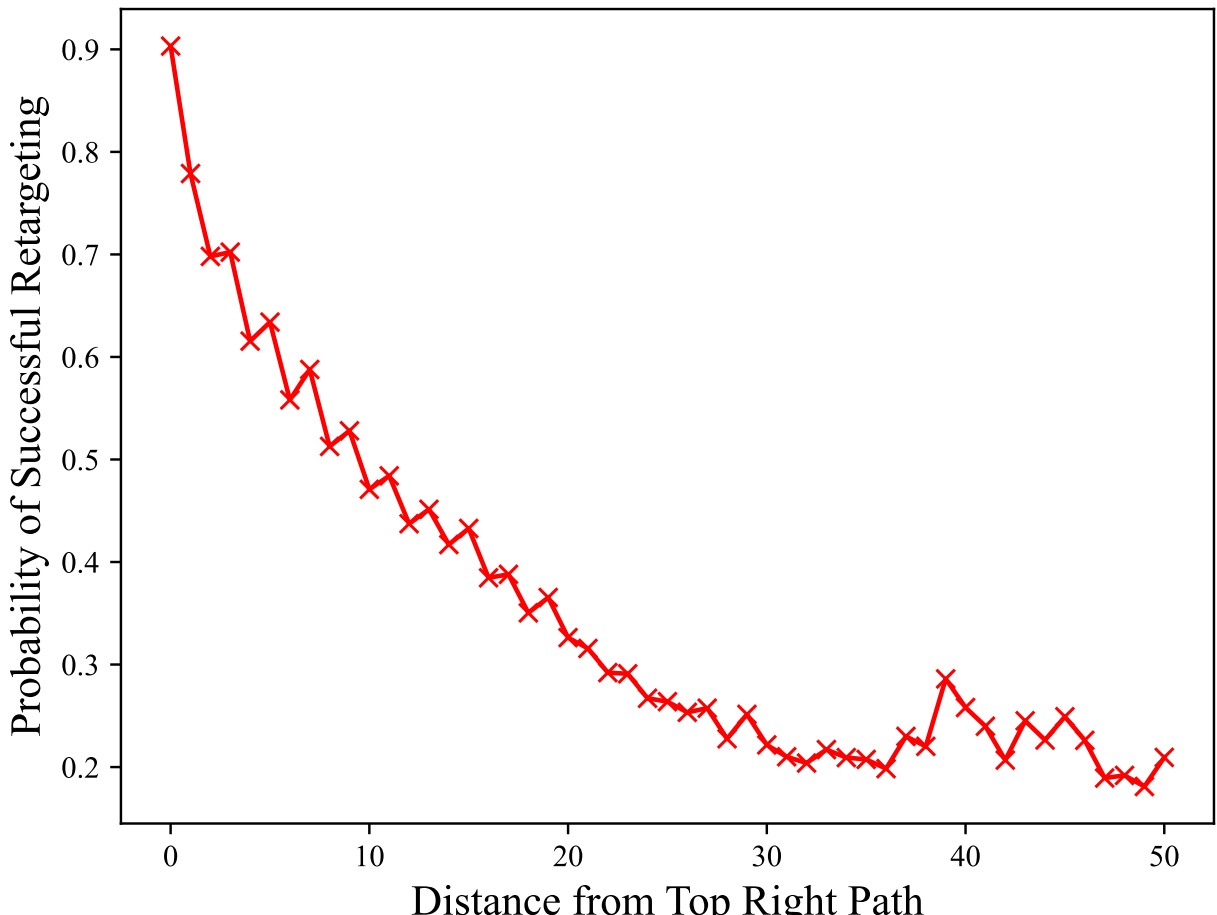

Figure 30

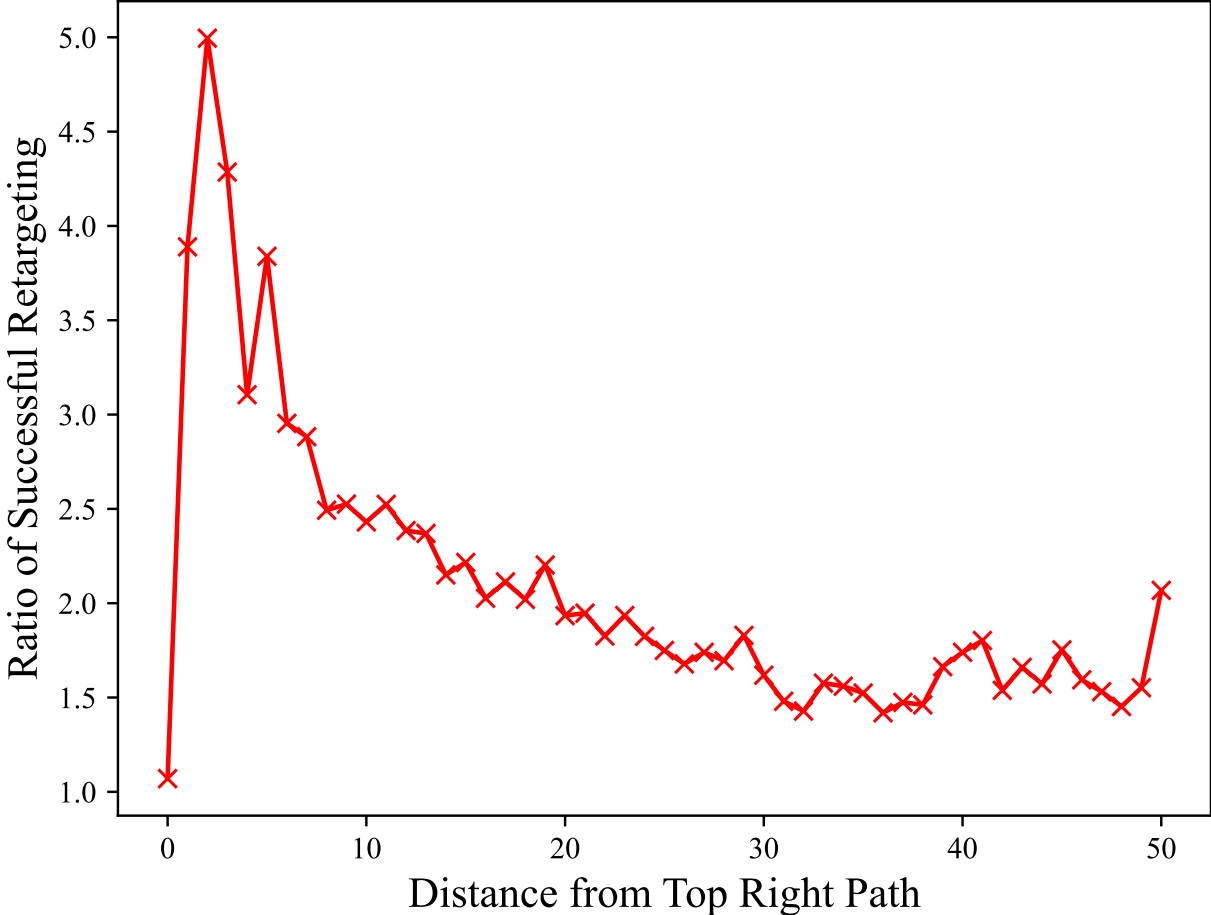

Figure 31: Targets $s_t$ which are farther off the top-right path, will often have lower retargeting probability *and* lower base probability. To control for this, we plot the *ratio* $\frac{P_{\text{path}}(s_t \mid \pi_{\texttt{retarget}})}{P_{\text{path}}(s_t \mid \pi)}$. Ratios greater than 1 indicate that the retargeting increased the normalized path probability. Thus, retargeting increases the probability of reaching a tile, no matter its distance from the top-right path.

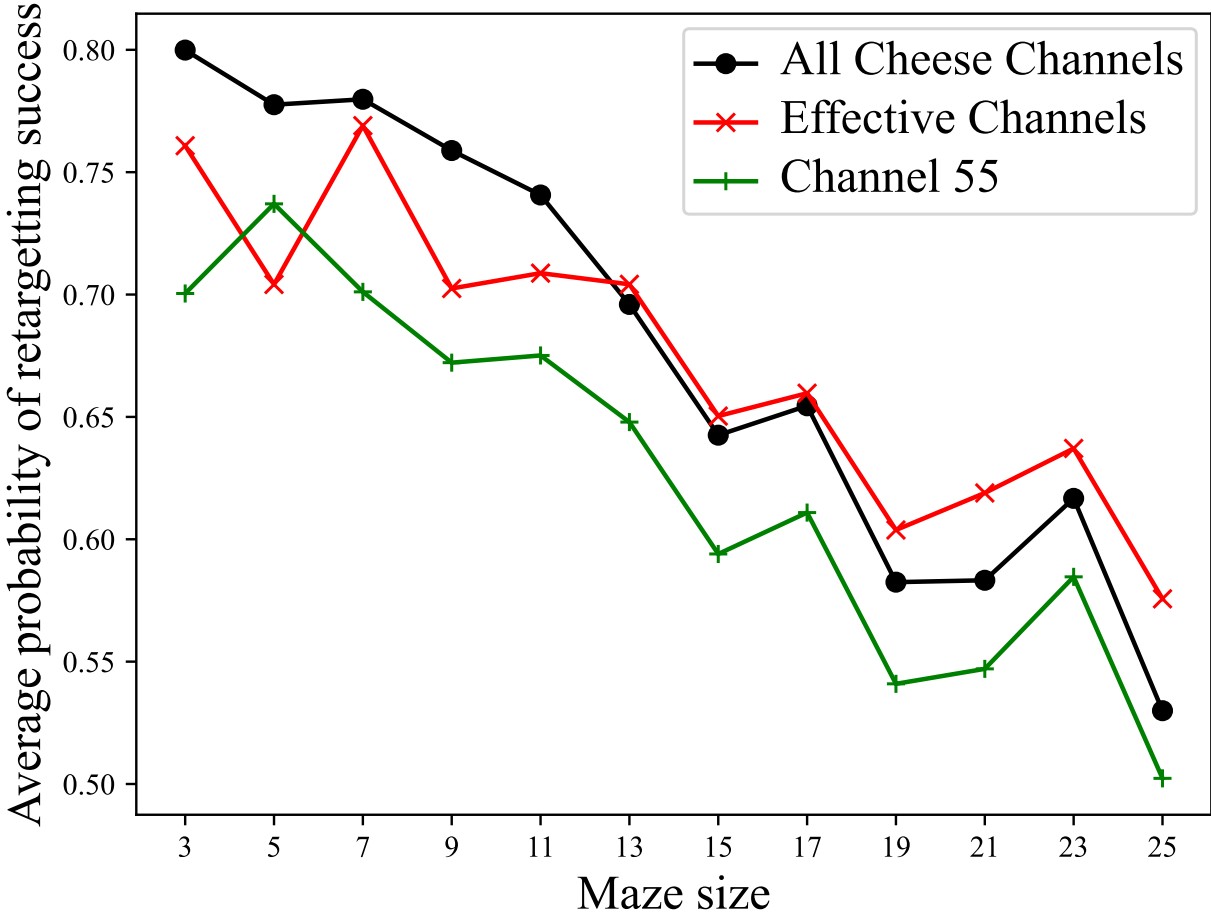

Figure 32: Modifying more channels increases the probability of successful retargeting over the whole maze in comparison to modifying a single channel. Crucially, this is true *even with larger magnitudes in the single-channel interventions*. This data shows that intervening on more of the circuits distributed throughout the network is more effective. Finding such circuits is important for controlling networks through manual activation engineering.

# E  Further Examples of Network Behavior

## E.1  Further Examples of fig. 3: *Network Channels Track The Goal Location*

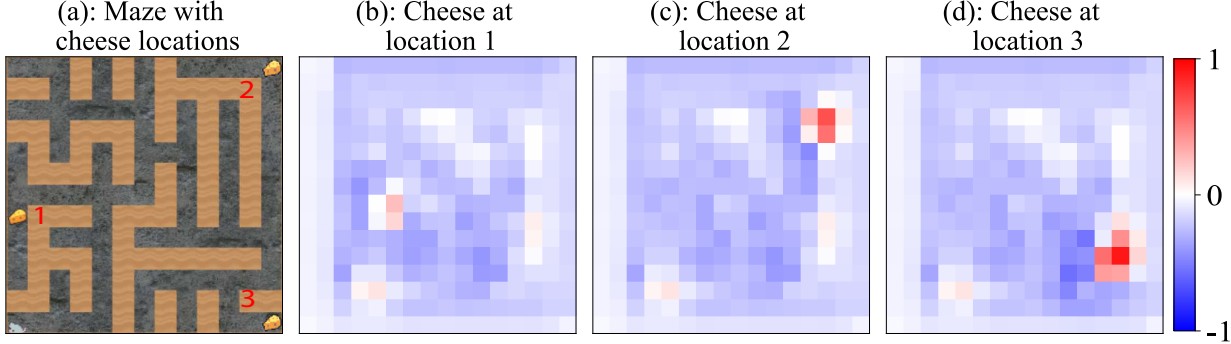

Figure 33: Channel 7.

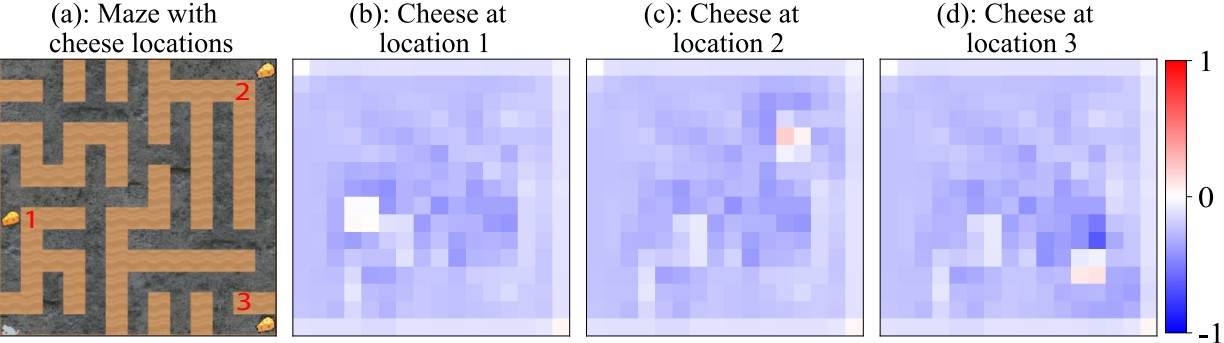

Figure 34: Channel 8.

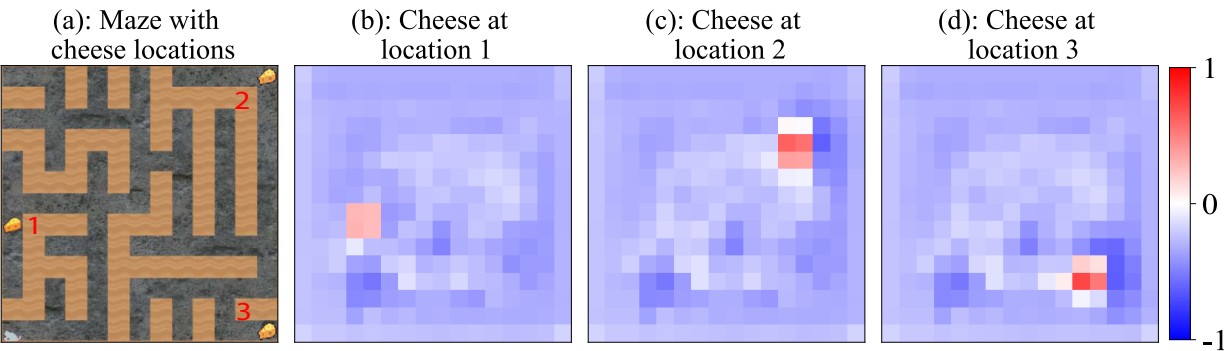

Figure 35: Channel 42.

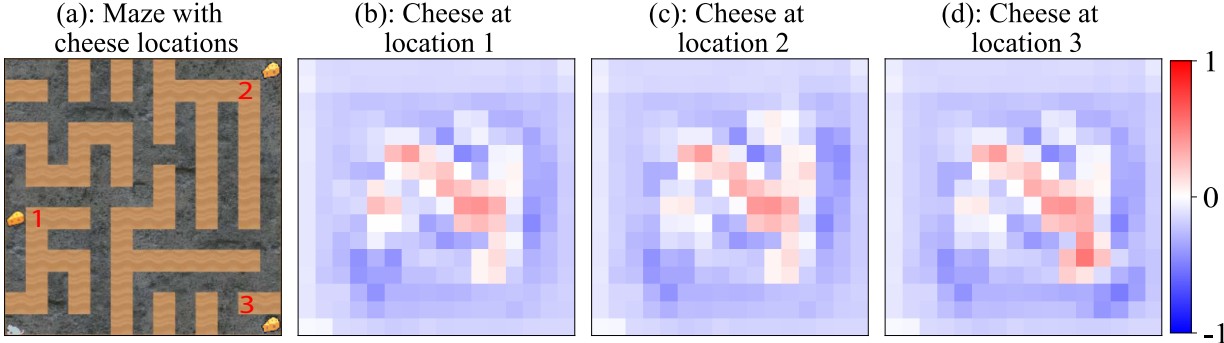

Figure 36: Channel 44.

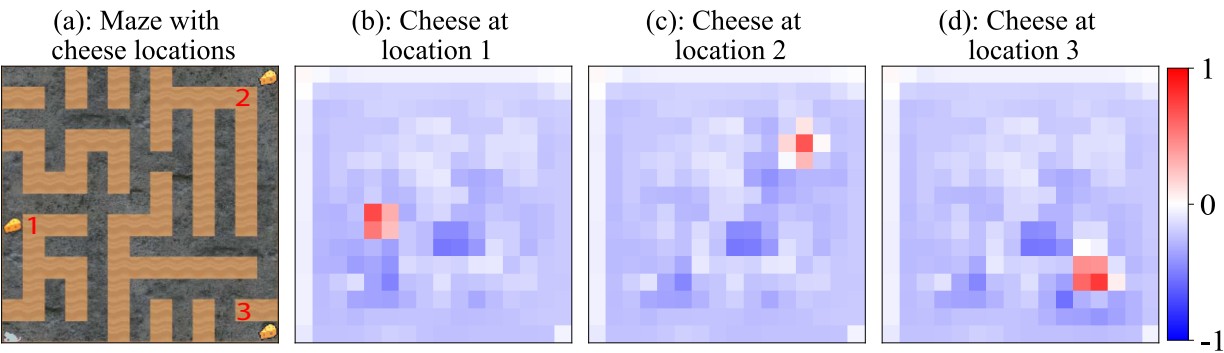

Figure 37: Channel 55.

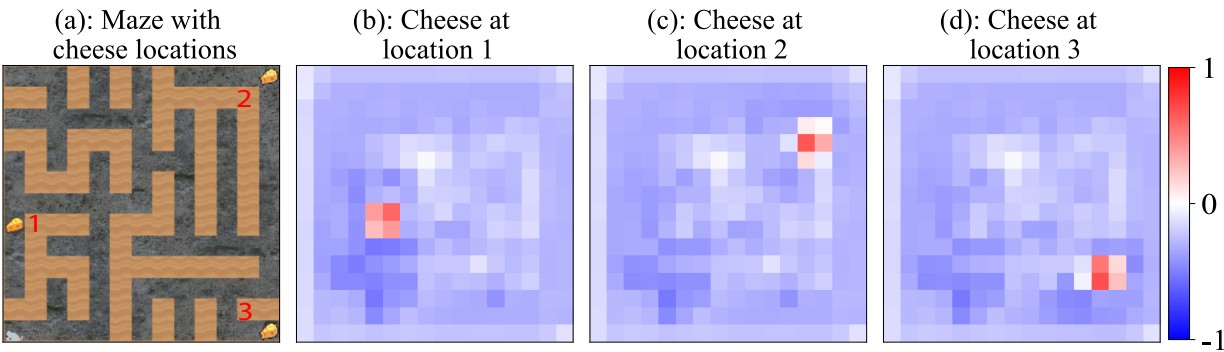

Figure 38: Channel 77.

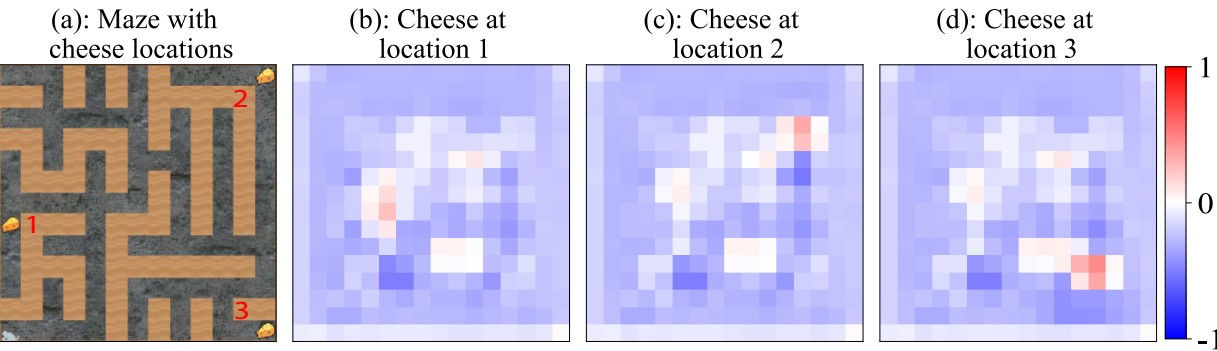

Figure 39: Channel 82.

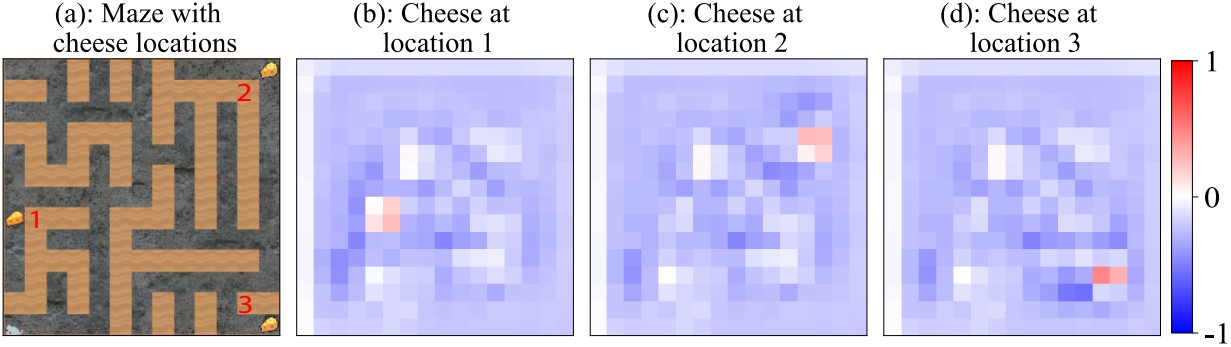

Figure 40: Channel 88.

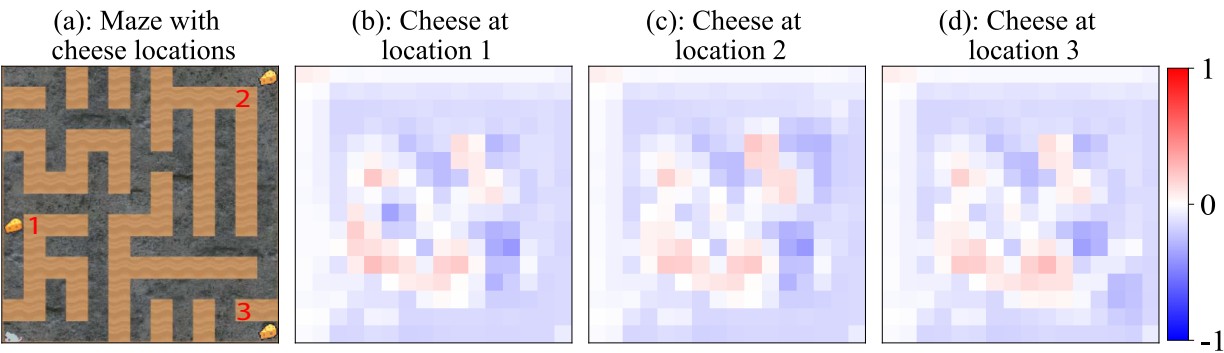

Figure 41: Channel 89.

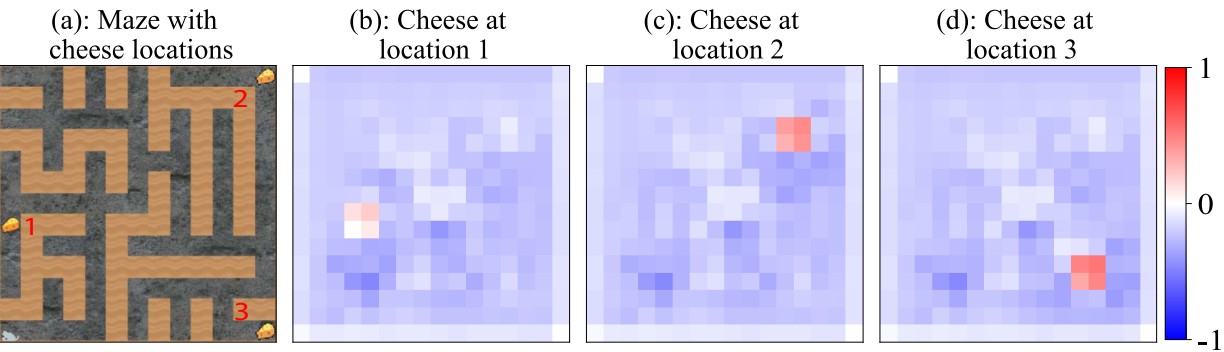

Figure 42: Channel 99.

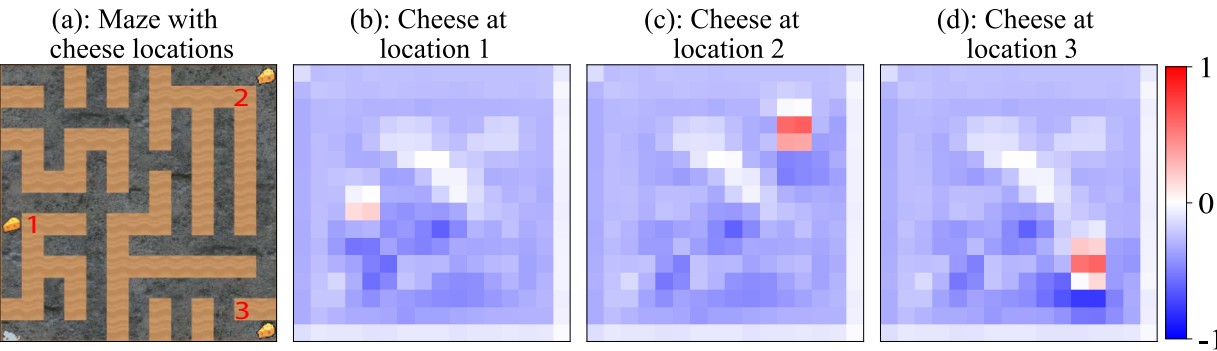

Figure 43: Channel 113.

### E.2 Further Examples of fig. 4: *Resampling Cheese-Tracking Activations From Different Mazes*

Here we show 3 examples of each size maze from the first 100 seeds. The resampled locations are always in the top right and bottom right corners, respectively. Resampling locations that were farther from the path to the top-right corner (appendix D for further details) were more difficult to steer towards. In most instances of resampling from cheese located in the bottom right, the policy instead steered towards the historical goal location in the top right.

(a): Original Maze

(b): Resampling from same location

(c): Resampling from red cheese location

(d): Resampling from red cheese location

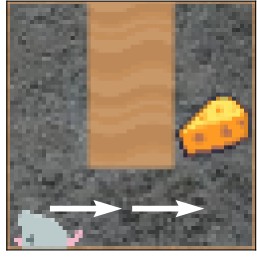 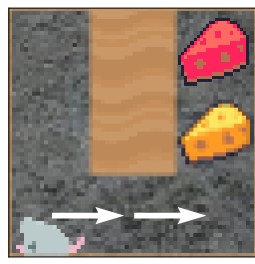 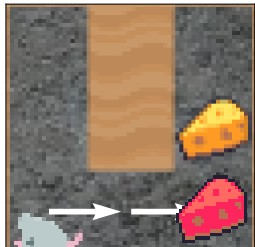

Figure 44: Maze size 3x3: seed 1.

(a): Original Maze

(b): Resampling from same location

(c): Resampling from red cheese location

(d): Resampling from red cheese location

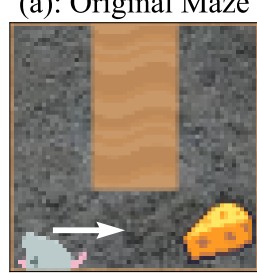 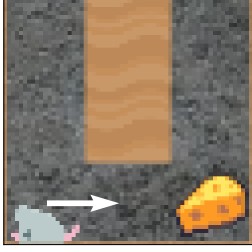 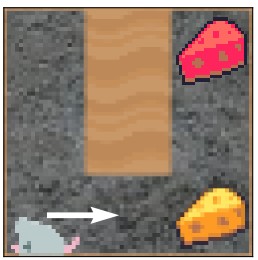 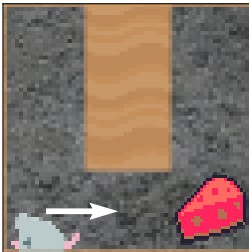

Figure 45: Maze size 3x3: seed 10.

(b): Resampling from same location

(c): Resampling from red cheese location

(d): Resampling from red cheese location

(a): Original Maze

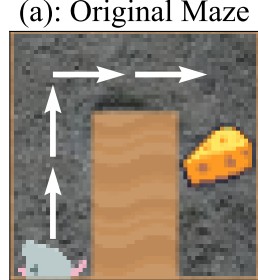 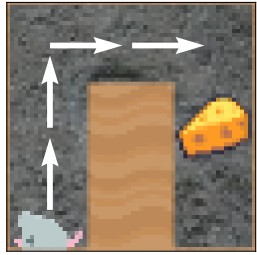 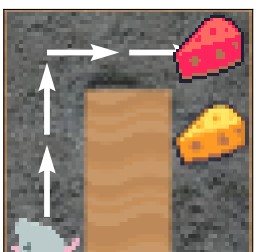 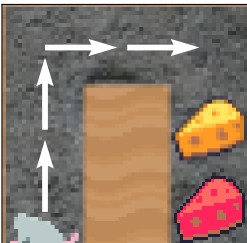

Figure 46: Maze size 3x3: seed 11.

(a): Original Maze | (b): Resampling from same location | (c): Resampling from red cheese location | (d): Resampling from red cheese location

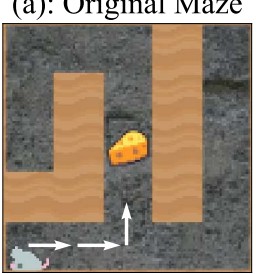 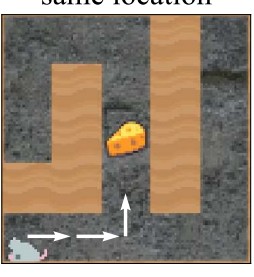 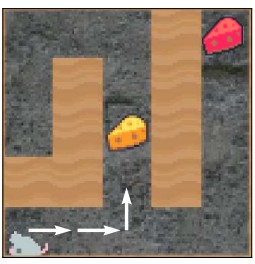 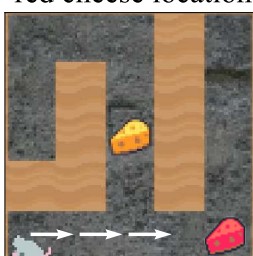

Figure 47: Maze size 5x5: seed 3.

(a): Original Maze | (b): Resampling from same location | (c): Resampling from red cheese location | (d): Resampling from red cheese location

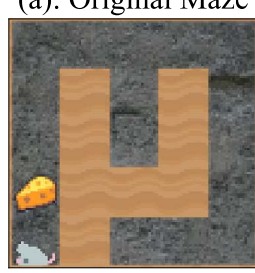 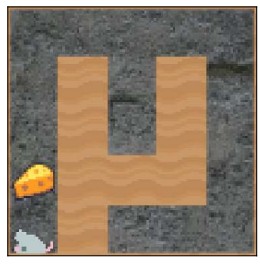 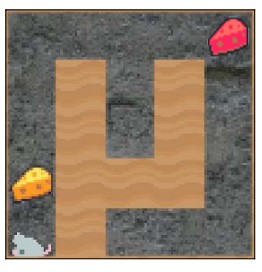 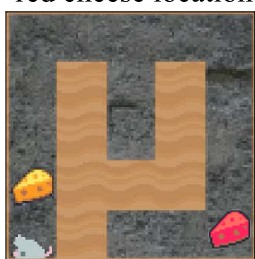

Figure 48: Maze size 5x5: seed 7.

(a): Original Maze | (b): Resampling from same location | (c): Resampling from red cheese location | (d): Resampling from red cheese location

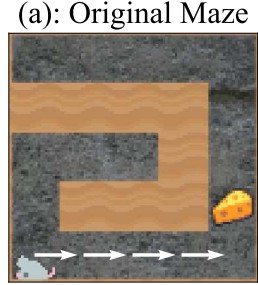 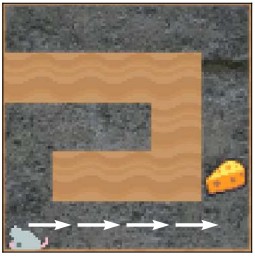 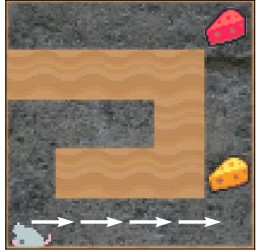 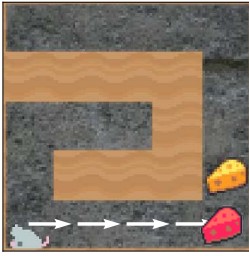

Figure 49: Maze size 5x5: seed 19.

(a): Original Maze | (b): Resampling from same location | (c): Resampling from red cheese location | (d): Resampling from red cheese location

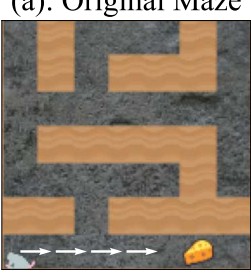 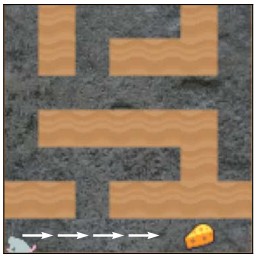 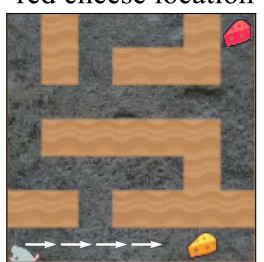 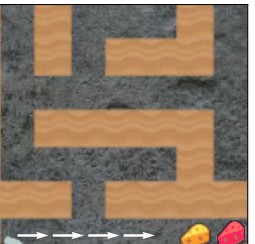

Figure 50: Maze size 7x7: seed 26.

(a): Original Maze

(b): Resampling from same location

(c): Resampling from red cheese location

(d): Resampling from red cheese location

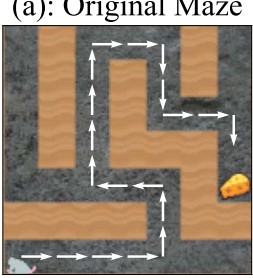 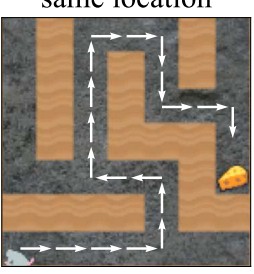 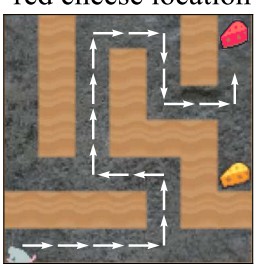 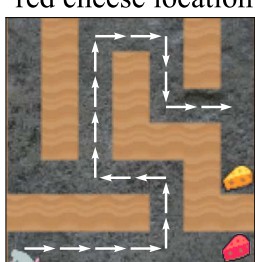

Figure 51: Maze size 7x7: seed 34.

(a): Original Maze

(b): Resampling from same location

(c): Resampling from red cheese location

(d): Resampling from red cheese location

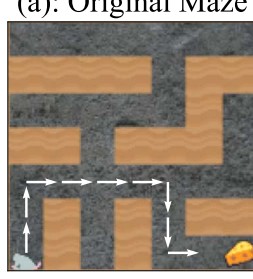 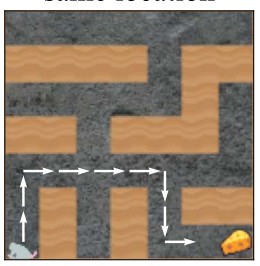 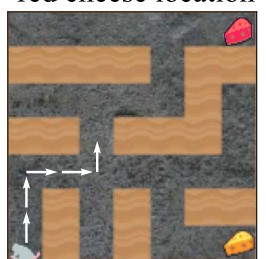 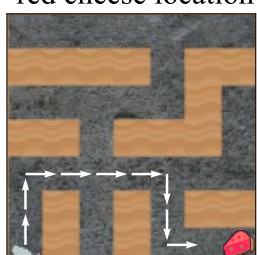

Figure 52: Maze size 7x7: seed 54.

(a): Original Maze

(b): Resampling from same location

(c): Resampling from red cheese location

(d): Resampling from red cheese location

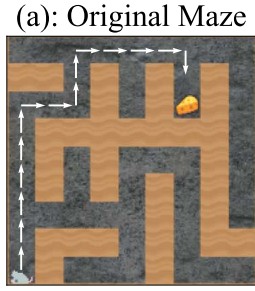 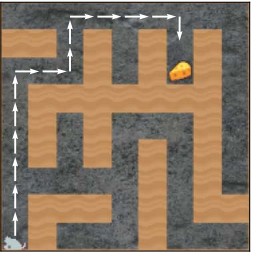 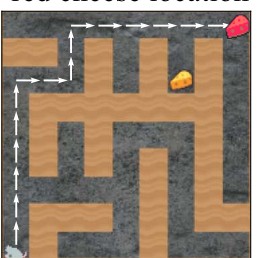 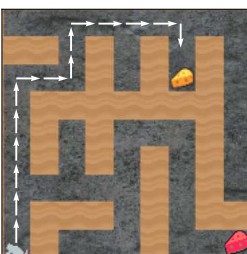

Figure 53: Maze size 7x7: seed 6.

(a): Original Maze

(b): Resampling from same location

(c): Resampling from red cheese location

(d): Resampling from red cheese location

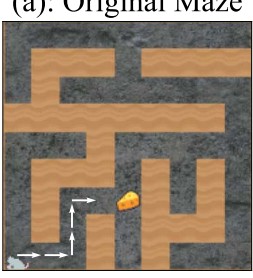 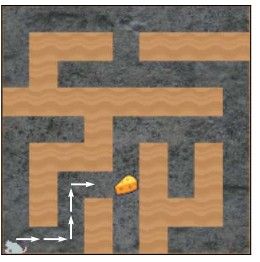 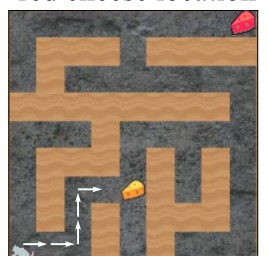 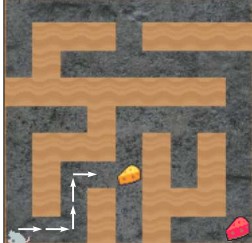

Figure 54: Maze size 7x7: seed 20.

(a): Original Maze

(b): Resampling from same location

(c): Resampling from red cheese location

(d): Resampling from red cheese location

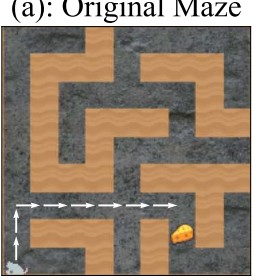
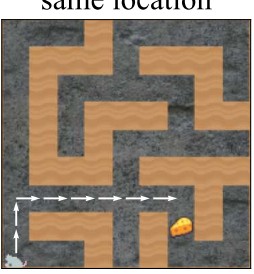
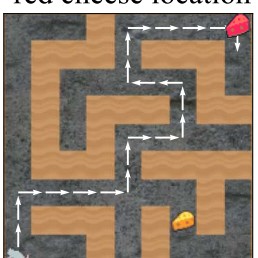
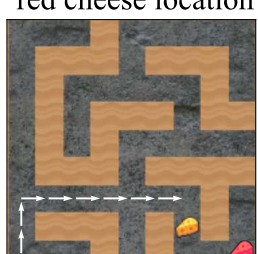

Figure 55: Maze size 7x7: seed 52.

(a): Original Maze

(b): Resampling from same location

(c): Resampling from red cheese location

(d): Resampling from red cheese location

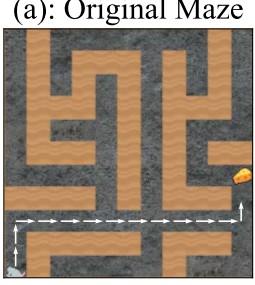
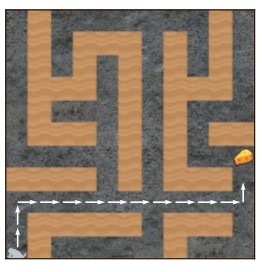
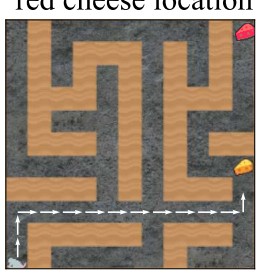
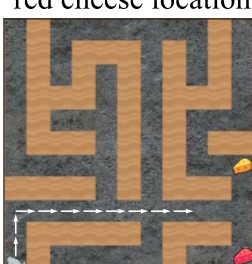

Figure 56: Maze size 11x11: seed 35.

(a): Original Maze

(b): Resampling from same location

(c): Resampling from red cheese location

(d): Resampling from red cheese location

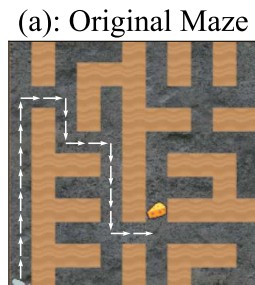
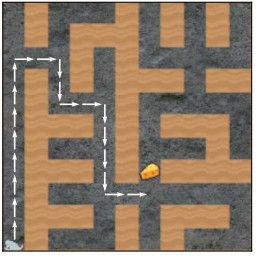
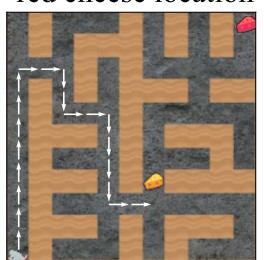
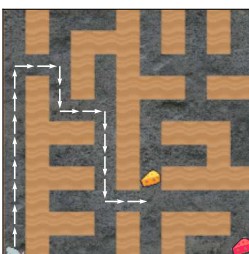

Figure 57: Maze size 11x11: seed 37.

(a): Original Maze

(b): Resampling from same location

(c): Resampling from red cheese location

(d): Resampling from red cheese location

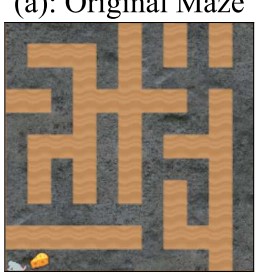
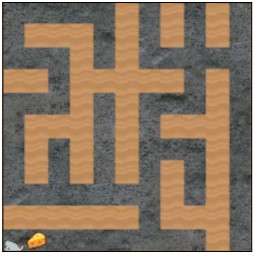
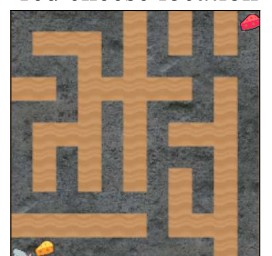
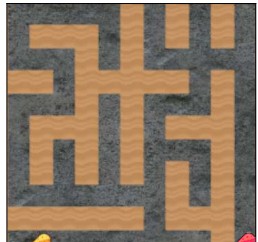

Figure 58: Maze size 11x11: seed 42.

(a): Original Maze

(b): Resampling from same location

(c): Resampling from red cheese location

(d): Resampling from red cheese location

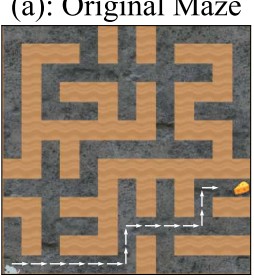
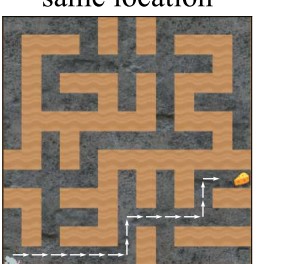
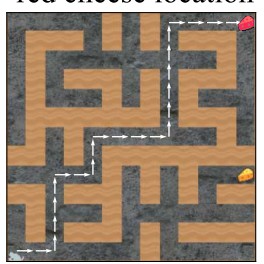
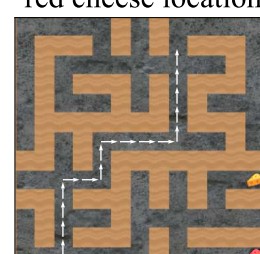

Figure 59: Maze size 13x13: seed 51.

(a): Original Maze

(b): Resampling from same location

(c): Resampling from red cheese location

(d): Resampling from red cheese location

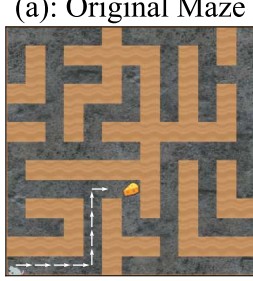
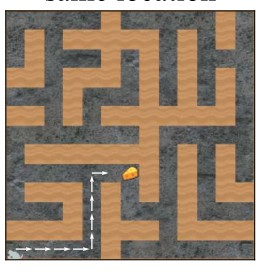
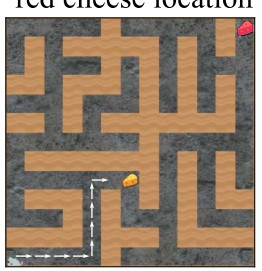
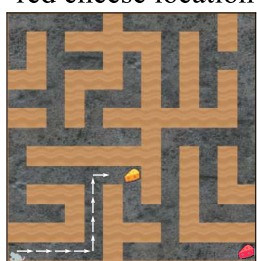

Figure 60: Maze size 13x13: seed 74.

(a): Original Maze

(b): Resampling from same location

(c): Resampling from red cheese location

(d): Resampling from red cheese location

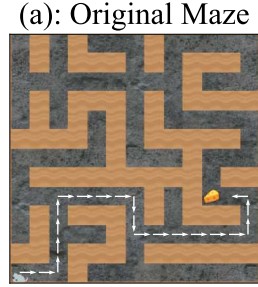
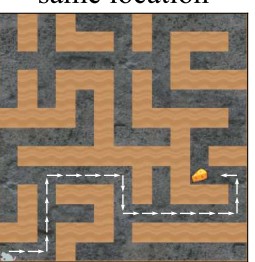
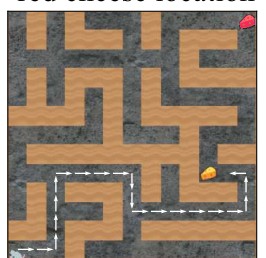
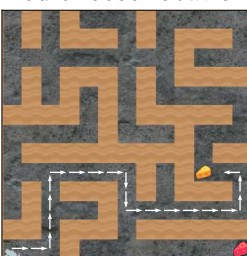

Figure 61: Maze size 13x13: seed 84.

(a): Original Maze

(b): Resampling from same location

(c): Resampling from red cheese location

(d): Resampling from red cheese location

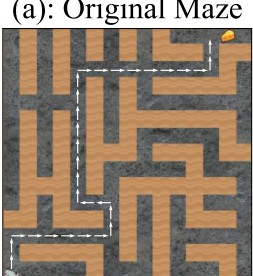
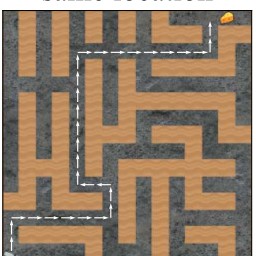
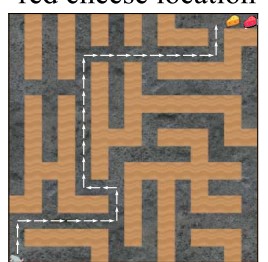
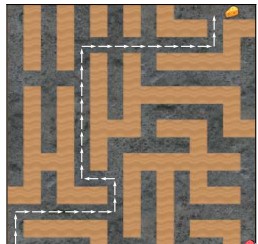

Figure 62: Maze size 15x15: seed 9.

(a): Original Maze

(b): Resampling from same location

(c): Resampling from red cheese location

(d): Resampling from red cheese location

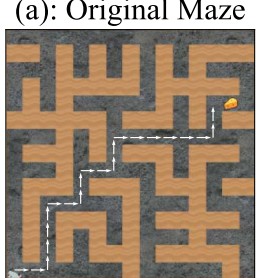
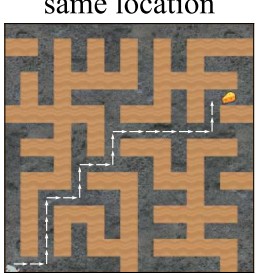
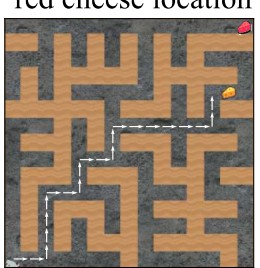
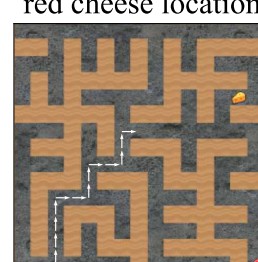

Figure 63: Maze size 15x15: seed 25.

(a): Original Maze

(b): Resampling from same location

(c): Resampling from red cheese location

(d): Resampling from red cheese location

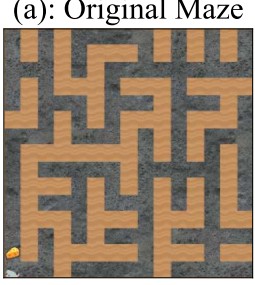
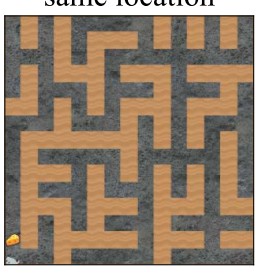
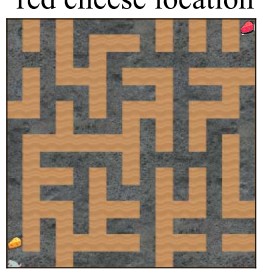
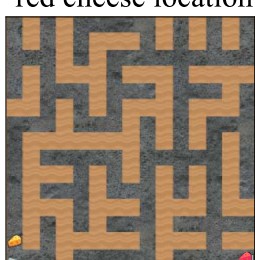

Figure 64: Maze size 15x15: seed 36.

(a): Original Maze

(b): Resampling from same location

(c): Resampling from red cheese location

(d): Resampling from red cheese location

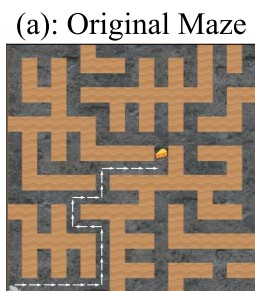
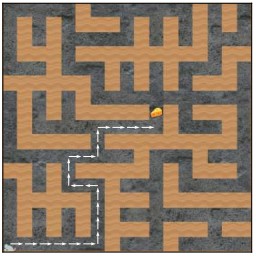
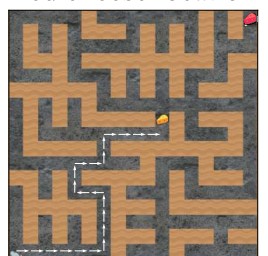
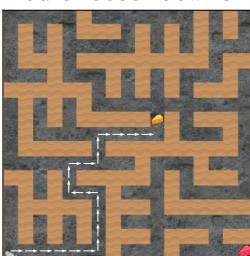

Figure 65: Maze size 17x17: seed 50.

(a): Original Maze

(b): Resampling from same location

(c): Resampling from red cheese location

(d): Resampling from red cheese location

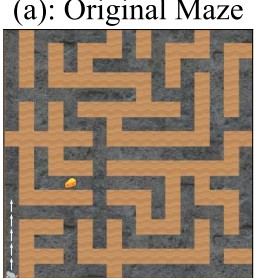
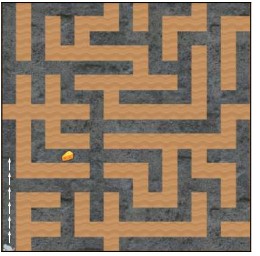
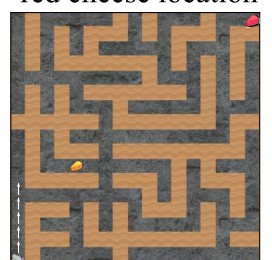
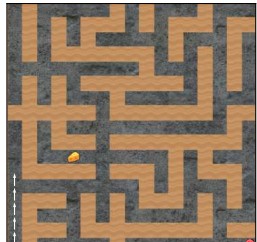

Figure 66: Maze size 17x17: seed 64.

(a): Original Maze

(b): Resampling from same location

(c): Resampling from red cheese location

(d): Resampling from red cheese location

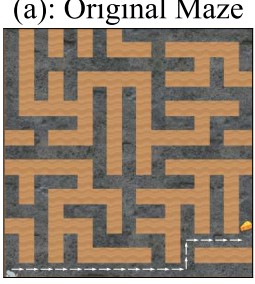
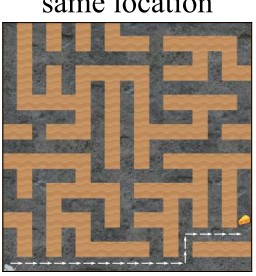
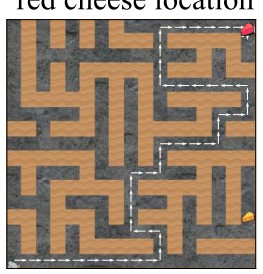
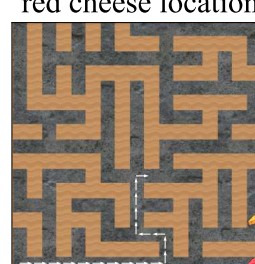

Figure 67: Maze size 17x17: seed 76.

(a): Original Maze

(b): Resampling from same location

(c): Resampling from red cheese location

(d): Resampling from red cheese location

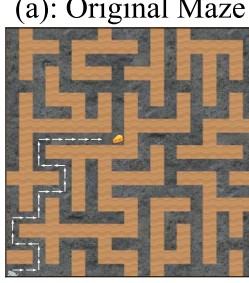
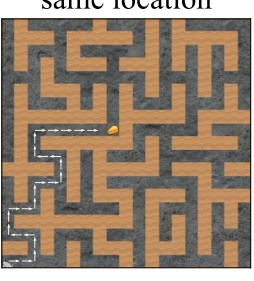
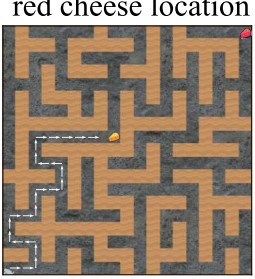
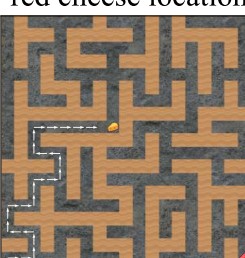

Figure 68: Maze size 19x19: seed 24.

(a): Original Maze

(b): Resampling from same location

(c): Resampling from red cheese location

(d): Resampling from red cheese location

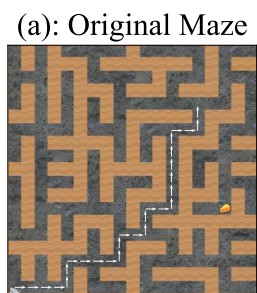
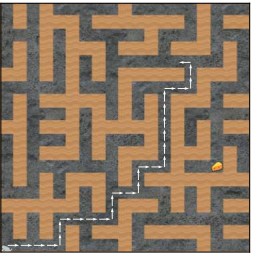
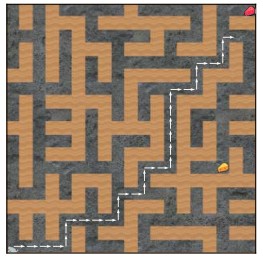
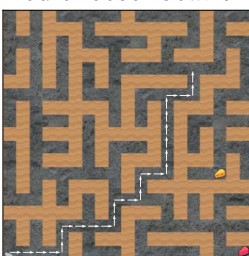

Figure 69: Maze size 19x19: seed 46.

(a): Original Maze

(b): Resampling from same location

(c): Resampling from red cheese location

(d): Resampling from red cheese location

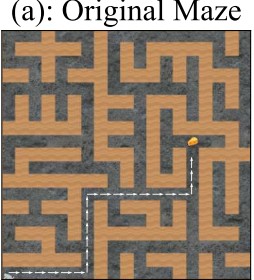
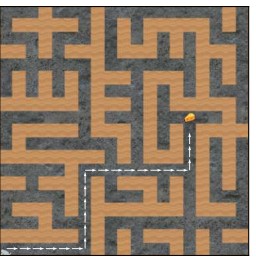
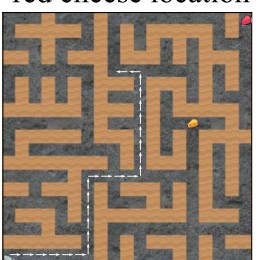
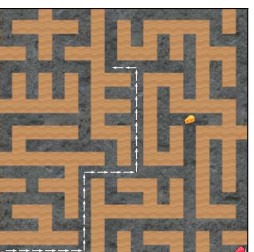

Figure 70: Maze size 19x19: seed 81.

(a): Original Maze

(b): Resampling from same location

(c): Resampling from red cheese location

(d): Resampling from red cheese location

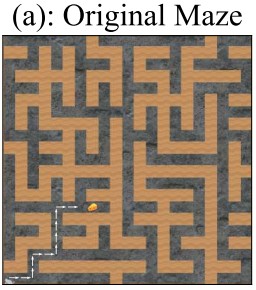 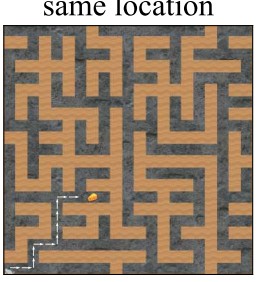 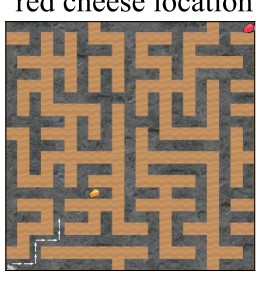 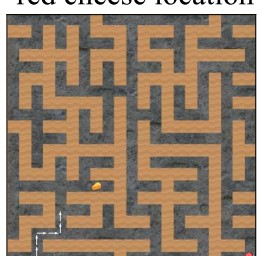

Figure 71: Maze size 21x21: seed 8.

(a): Original Maze

(b): Resampling from same location

(c): Resampling from red cheese location

(d): Resampling from red cheese location

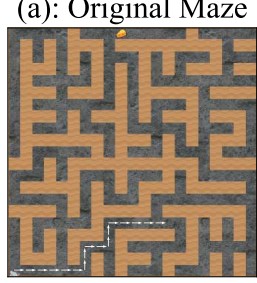 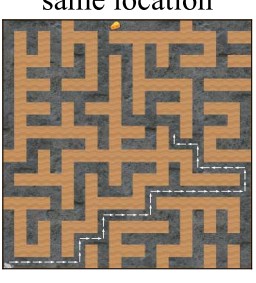 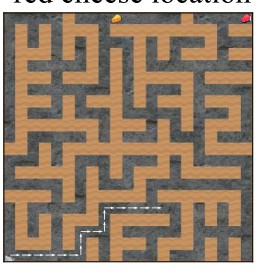 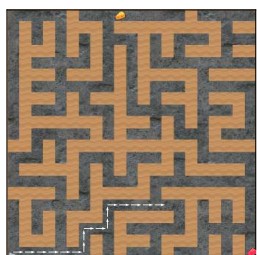

Figure 72: Maze size 21x21: seed 32.

(a): Original Maze

(b): Resampling from same location

(c): Resampling from red cheese location

(d): Resampling from red cheese location

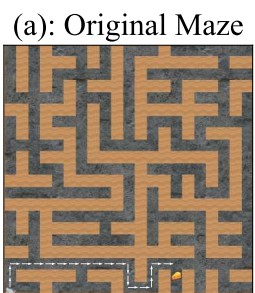 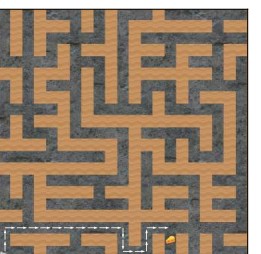 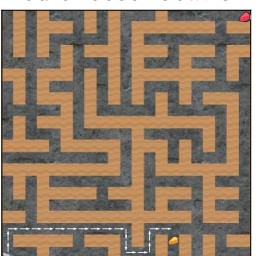 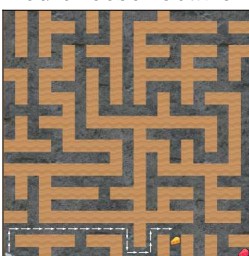

Figure 73: Maze size 21x21: seed 53.

(a): Original Maze

(b): Resampling from same location

(c): Resampling from red cheese location

(d): Resampling from red cheese location

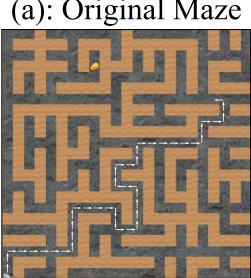 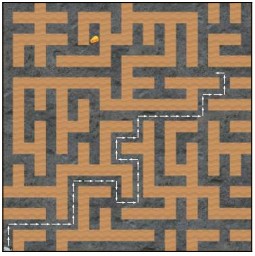 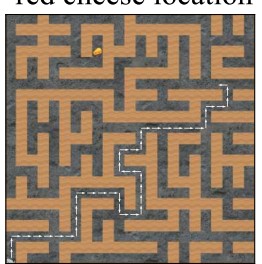 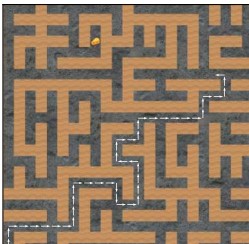

Figure 74: Maze size 23x23: seed 12.

(a): Original Maze

(b): Resampling from same location

(c): Resampling from red cheese location

(d): Resampling from red cheese location

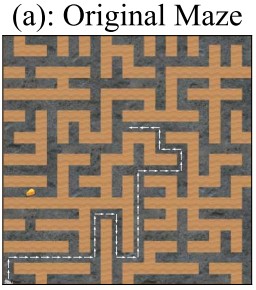 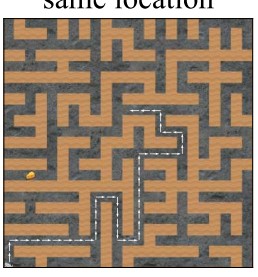 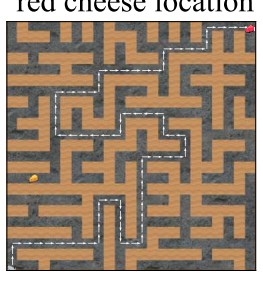 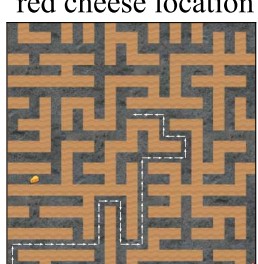

Figure 75: Maze size 23x23: seed 13.

(a): Original Maze

(b): Resampling from same location

(c): Resampling from red cheese location

(d): Resampling from red cheese location

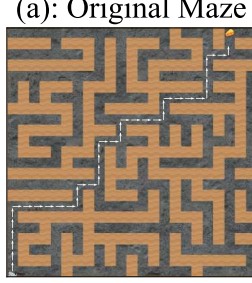 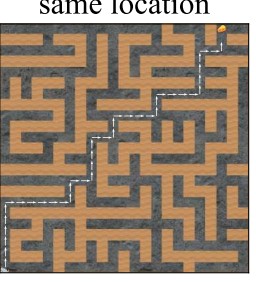 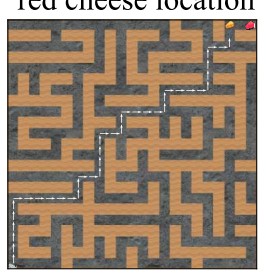 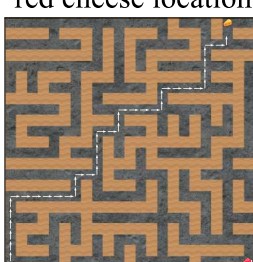

Figure 76: Maze size 23x23: seed 67.

(a): Original Maze

(b): Resampling from same location

(c): Resampling from red cheese location

(d): Resampling from red cheese location

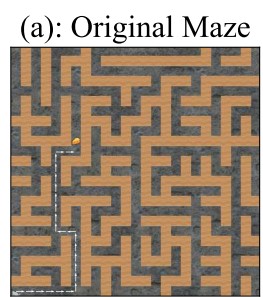 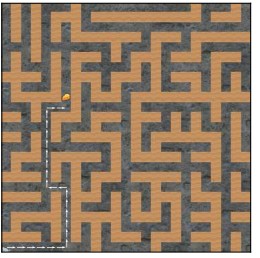 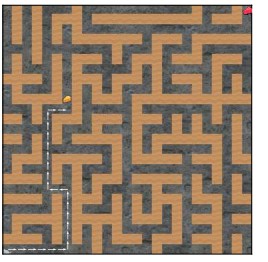 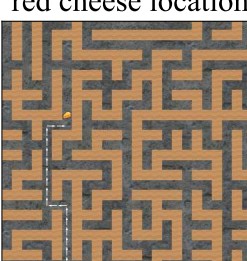

Figure 77: Maze size 25x25: seed 40.

(a): Original Maze

(b): Resampling from same location

(c): Resampling from red cheese location

(d): Resampling from red cheese location

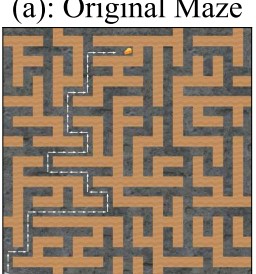 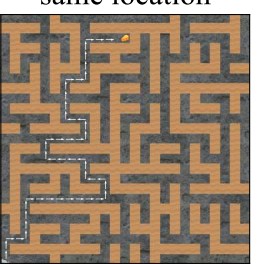 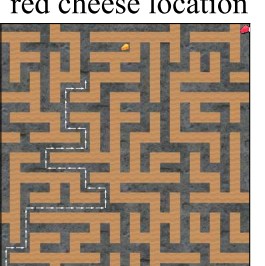 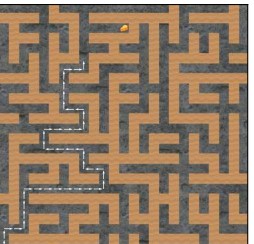

Figure 78: Maze size 25x25: seed 55.

(a): Original Maze

(b): Resampling from same location

(c): Resampling from red cheese location

(d): Resampling from red cheese location

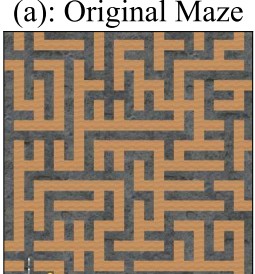 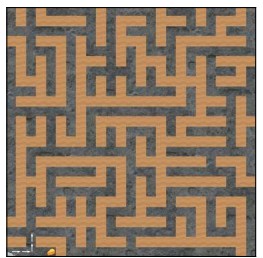 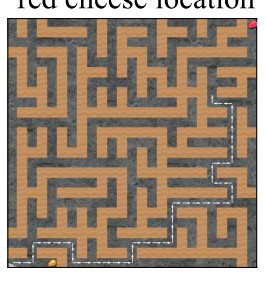 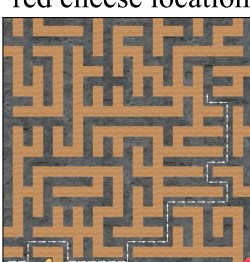

Figure 79: Maze size 25x25: seed 71.

### E.3 Further Examples of fig. 5: *Controlling The Maze-Solving Policy By Modifying A Single Activation*

Here we take the same specific activations from fig. 5, with intervention magnitude $\alpha = +5.5$, and apply them to other mazes of the same size. Arbitrary retargeting does not always work, especially for activations farther away from the top-right path. See appendix D for more information and statistics on the top-right path. The most-probable paths indicate that it's harder to retarget the mouse farther off of the top-right path. Instead, the policy navigates to the historical goal location. In fact, some seeds do not see any change in the most probable path, although quantitative analyses in appendix D detail the changing probabilities of all paths through different maze sizes and interventions.

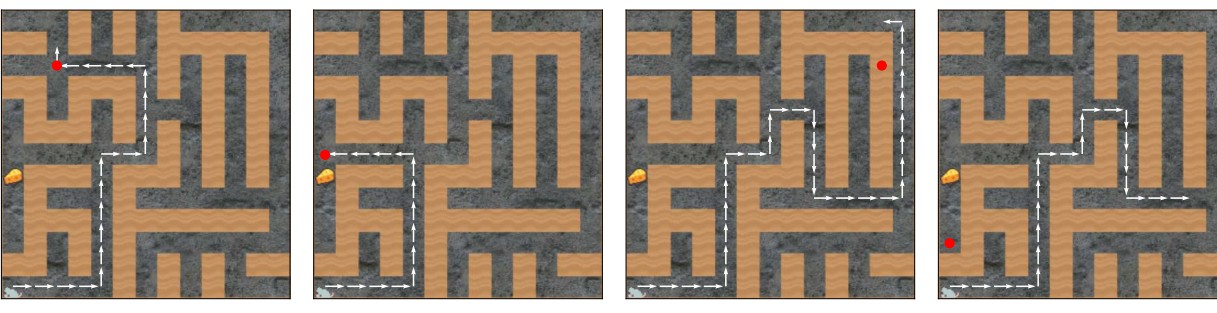

Figure 80: Patching specific activations: seed 0.

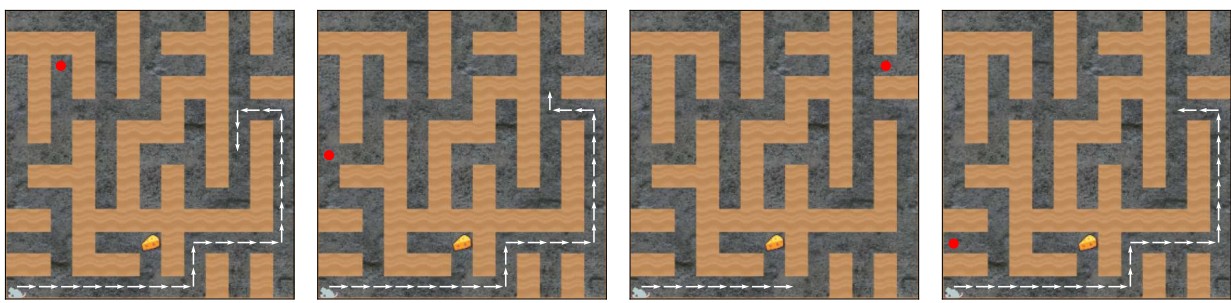

Figure 81: Patching specific activations: seed 2.

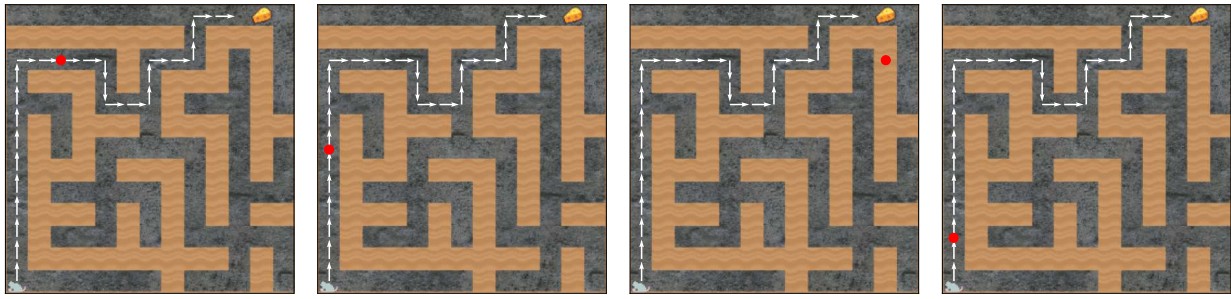

Figure 82: Patching specific activations: seed 16.

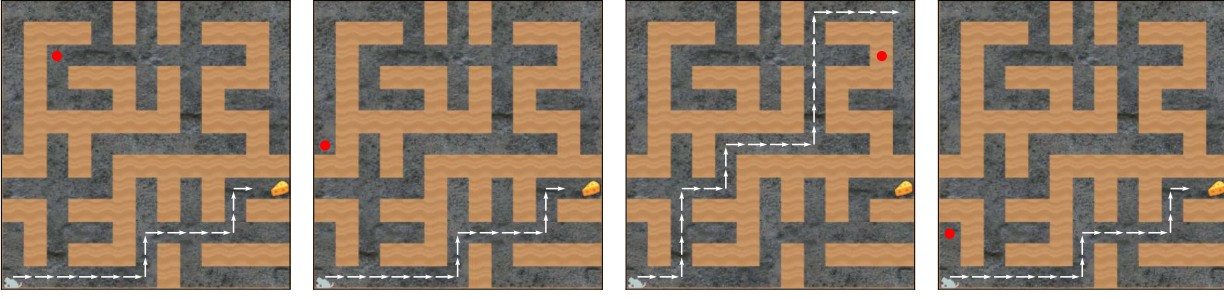

Figure 83: Patching specific activations: seed 51.

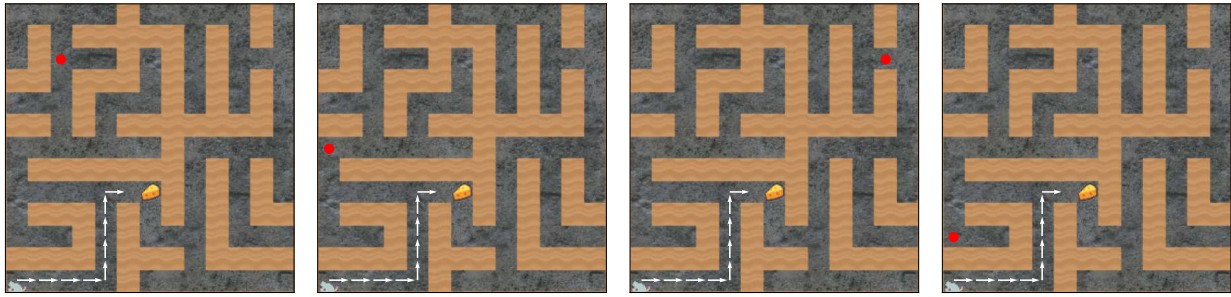

Figure 84: Patching specific activations: seed 74.

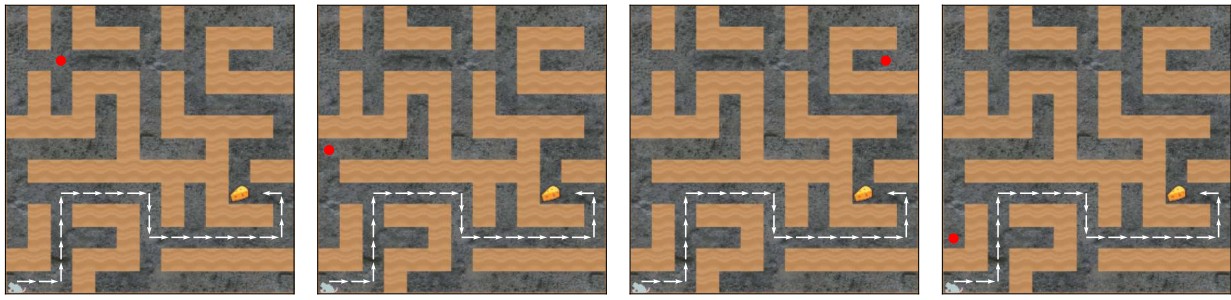

Figure 85: Patching specific activations: seed 84.

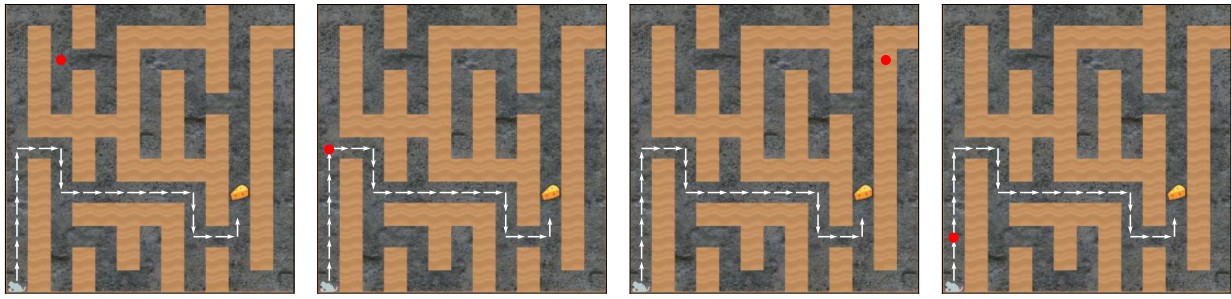

Figure 86: Patching specific activations: seed 85.

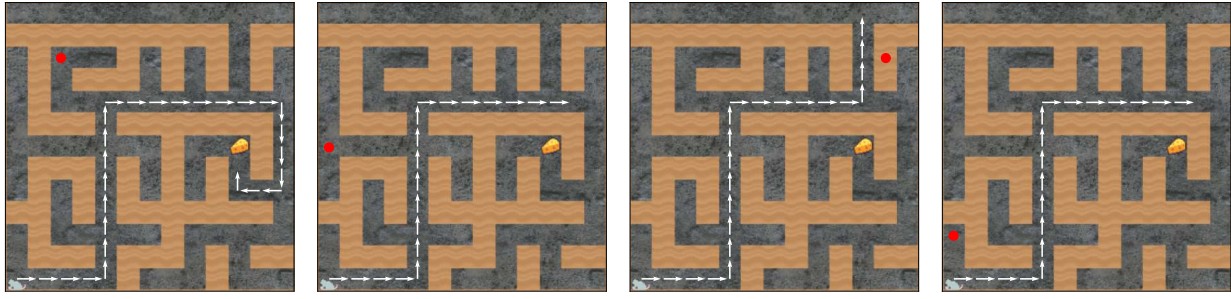

Figure 87: Patching specific activations: seed 99.

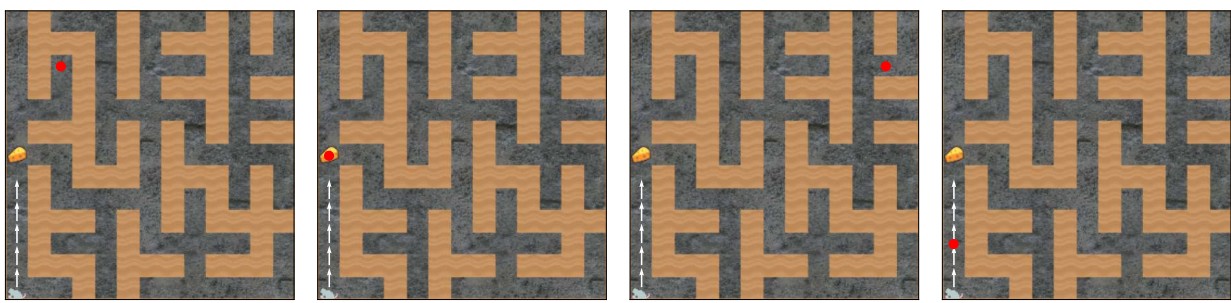

Figure 88: Patching specific activations: seed 107.

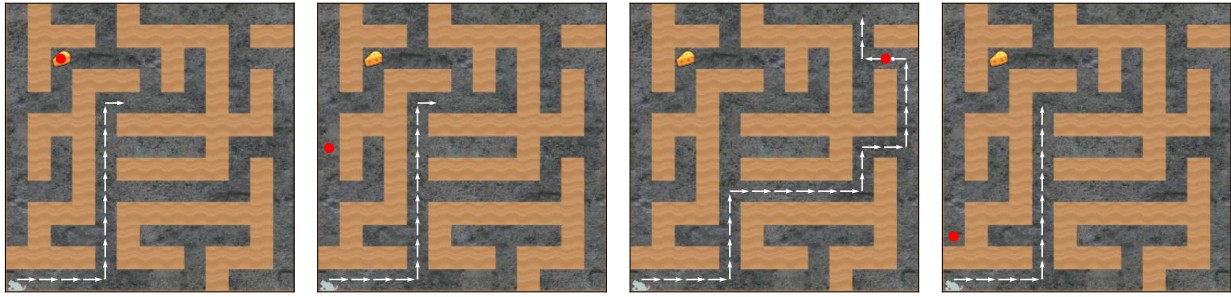

Figure 89: Patching specific activations: seed 108.

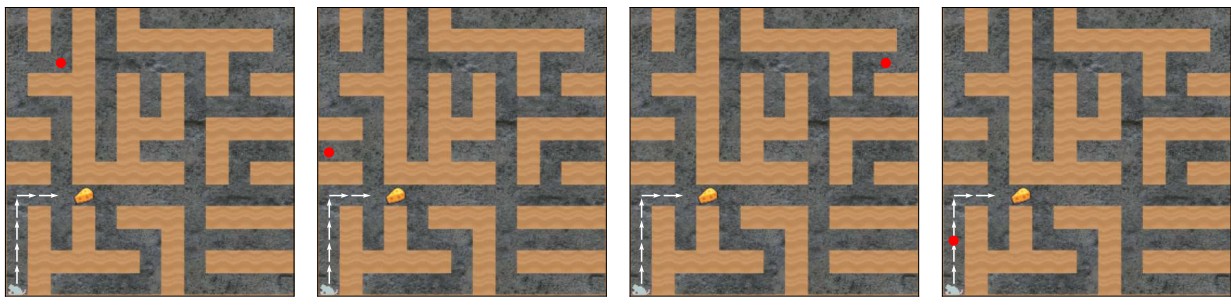

Figure 90: Patching specific activations: seed 132.

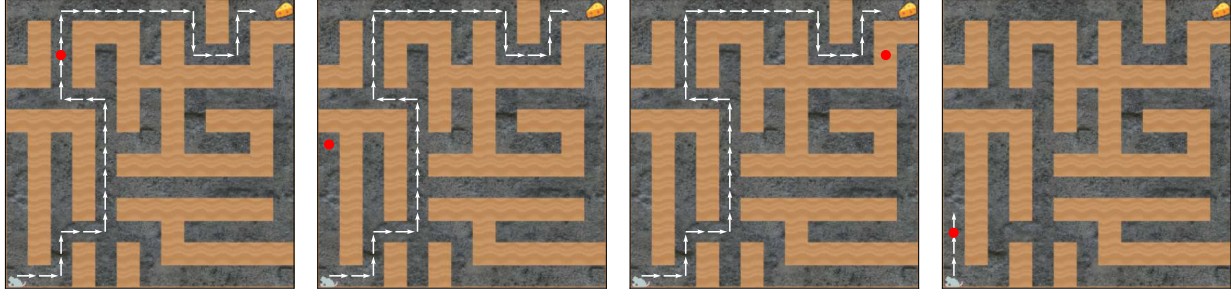

Figure 91: Patching specific activations: seed 169.

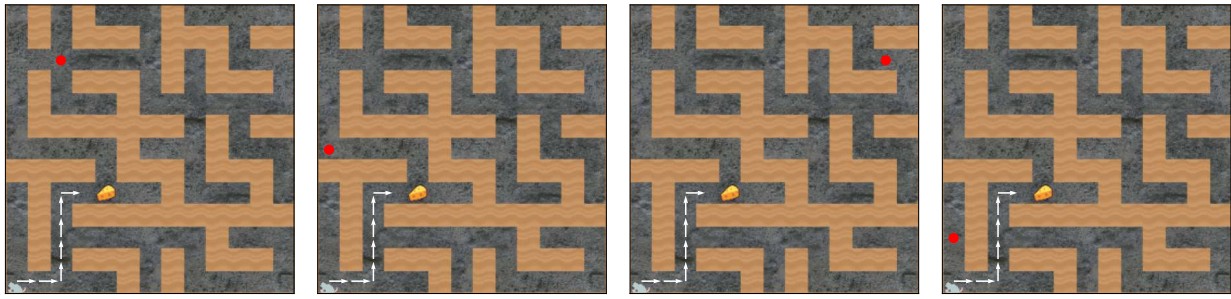

Figure 92: Patching specific activations: seed 183.

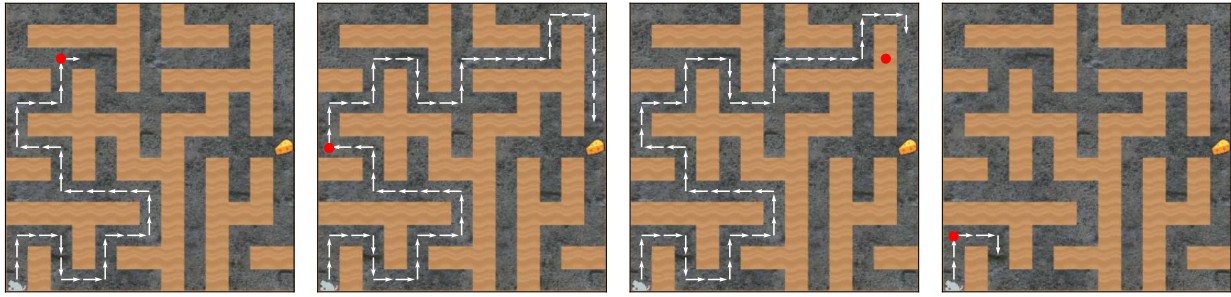

Figure 93: Patching specific activations: seed 189.

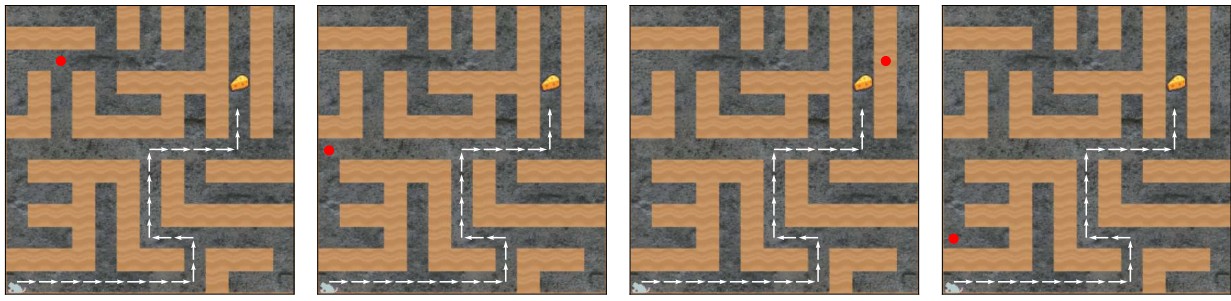

Figure 94: Patching specific activations: seed 192.

