# OpenReview forum: "Understanding and Controlling a Maze-solving Policy Network"
_TMLR — Rejected by TMLR_

### Review · Reviewer_YzDZ · 2024-10-09

**Summary Of Contributions:**

The paper is about understanding and controlling a maze-solving policy network through the study of its goals and goal representations in AI systems. A pre-trained reinforcement learning policy solving mazes reveals multiple context-dependent goals, with eleven channels tracking goal location. By modifying these channels, policy control can be partially achieved. The network contains redundant, distributed, and retargetable goal representations, illuminating goal-direction in trained policy networks.

**Audience:**

No

**Broader Impact Concerns:**

No ethical implications in the work

**Claims And Evidence:**

No

**Requested Changes:**

- The paper needs to be re-written to improve English and readability
- The discussion must be generalized to be applied to other contexts

**Strengths And Weaknesses:**

Weaknesses:

- The paper only refers to a specific toy example without practical application
- No discussion on how to generalize the idea to other contexts is given
- The discussion (section 6) fails to summarize the paper and its findings
- No discussion about the results obtained in the paper by Langosco et al. 2023 is provided. Does their model solve the problem? How?

---

> ### Author Response · Authors · 2024-11-11
> **Response to YzDZ**
>
> Thank you for your review. We understand that you have several concerns about our paper. First, we agree that we should give more context to our results and potential future work – we will add this to the camera-ready. However, we overall stand by our work and think it tackles an interesting problem in a sound manner (see Z5g8’s review).
>
> **Responses to other concerns.** You claim that we only examine a “toy example without practical application.” However, TMLR's acceptance criteria emphasize sound research and audience interest, not just practical application. [TMLR guidelines](https://jmlr.org/tmlr/editorial-policies.html) state that:
>
> > The acceptance decision for a submission is based on the answers to the following questions:
> > - _Are the claims made in the submission supported by accurate, convincing and clear evidence?_
> > - _Would at least some individuals in TMLR's audience be interested in knowing the findings of this paper?_
> >
> > Papers should be accepted if they meet the criteria, _even if the contribution or significance of the work is modest._
>
> Furthermore, even though we only perform a case study, the insights generalize. For example, the cheese vector technique has (via follow-up work) inspired dozens of papers analyzing LLM representations using vector steering approaches.
>
> > The discussion (section 6) fails to summarize the paper and its findings.
>
> The discussion (actually section 5, not 6) does in fact summarize the paper and its findings, though we plan to add more context and future work.
>
> > No discussion about the results obtained in the paper by Langosco et al. 2023 is provided. Does their model solve the problem? How?
>
> We find this confusing, as our paper is concerned with analyzing when and how their model “solves” the RL problem (of generalizing correctly to navigate to the cheese). Can you clarify?
>
> > The paper needs to be re-written to improve English and readability
>
> We want our paper to be easy to read and we welcome specific suggestions. However, this concern is too vague to implement as written – we aren’t sure what you found hard to read. We proofread the paper and sought feedback from colleagues before submitting. Reviewer Z5g8 stated that the paper is “well-written and structured, making it easy to follow the authors' line of reasoning, and the presentation is overall pleasant and relatively clear.” Perhaps you disagree, in which case we will eagerly improve any relevant areas you point out. Can you clarify?
>
> ----
>
> Overall, we think our paper meets the TMLR acceptance threshold. We look forward to working with you to further improve the paper and address your concerns.

---

### Review · Reviewer_Z5g8 · 2024-10-18

**Summary Of Contributions:**

This work studies the representation of goal states in an agent trained to navigate simple 2D procgen mazes. The authors do this by finding and manipulating internal circuits of a pretrained policy network, identifying the ones that track the location of goal states (e.g. cheese in the maze), and using these circuits to partially control the policy's behavior by manipulating the activations and by modifying forward passes with tensors containing information about the goal (or the state).

The manuscript proposes a set of methods for analyzing the behavior of the policy--largely straightforward regression methods combined with distance metrics in grid space. They find that they are able to segment the behavior of the target policy based on a subset of spatial features of the MDP.

**Audience:**

Yes

**Claims And Evidence:**

No

**Requested Changes:**

- [c1] I believe it is crucial to expand the mnuscript towards increasing the generalizability and significance of the methodology as well as the analysis. The authors might want to analyze more than one trained policy network and ideally include different tasks, architectures, or even training regimes. This is a major revision that would likely involve running new experiments and analyzing new data.

- [c2] Provide concrete examples of how the authors' methods could be used for alignment verification, safer exploration in RL, or correcting unwanted bias, in scenarios where these kinds of path finding networks are used (e.g. robotics?). This would increase the practical impact of the paper and make it more relevant to a broader audience of AI researchers that are interested in rationalizing and/or validating black box (neural) agents in this domain.

- [c3] I (and some other readers) would appreciate a more include a more comprehensive review of relevant work in interpretability in RL. The manuscript does a good job at discussing existing mechanistic intepretability literature, but work on analyzing navigation agents has been carried out for decades across both neuroscience venues and RL itself. I believe reframing the proposed method and experiments around what has been done in similar settings would be helpful to strengthen the work from both a methodology perspective as well as a scientific one.

**Strengths And Weaknesses:**

### Strengths

- [s1] The paper explores an intriguing question and understudied question. That is, how are goals represented in a trained policy network? The idea of studying AI systems like model organisms in biology is compelling, and there isn't nearly enough work in the literature that attempts to shed a light on how black-box (RL) agents behave mechanistically.

- [s2] The paper is well-written and structured, making it easy to follow the authors' line of reasoning, and the presentation is overall pleasant and relatively clear.

- [s3] The employed methodology is compelling. Circuit identification and manipulation has been used over the decades to carry out interpretability work, and it's a great way to empirically compose theories from behavioral data. Furthermore, I haven't seen much work manipulating the policy networks using feature vectors, and I can see the technique being very useful in all kinds of different work (especially given how straightforward it is).

### Weaknesses

- [w1] The paper's focus on a single, deep, convolutional policy network trained on a specific task (maze navigation) limits the generalizability of its findings. This makes the findings a little weak, and doesn't stretch nearly enough what the method could (and could not) do. The paper overall could be significantly strengthened by expanding the analysis to include other policy networks trained on different tasks or with different architectures. There are multiple dimensions that could be easily assessed against, even withing procgen itself (though I would suggest looking at multiple different benchmarks if possible).

- [w2] The statistical analysis of maze factors that predict the network's behavior, while suggestive, is relatively simple, and might not work as well in the wild (or in common  but more complex RL benchmarks--e.g. NetHack). For instance, using l1-regularized logistic regression as the only model might not be sufficient to capture the complex relationships between maze features and the network's behavior when the features of a goal state are less obviously qualitatively clear. I don't think it's unreasonable to assume that the method might not work particularly well in more complex agent-environment pairs.

- [w3] The authors briefly touch upon the potential applications of their findings, but they don't adequately explore _how_ the methods would broadly scale on these applications. RL policies tend to be nowadays used in highly dimensional (and critical!) environments that don't tend to be very amenable to this type of interpretability methods (e.g. https://arxiv.org/abs/1811.00260), and discussing these issues would be very valuable.

- [w4] The paper focuses heavily on identifying these "cheese-tracking" channels for the particular procgen environment. The analysis relies heavily on the fact that goals (or highly rewarding action-transitions) can be easily mapped to the cheese cells. There's a risk that the manuscript may be overinterpreting the role of these channels, and that this particular internal type of curcuits might arise only in these straightforward environments.

- [w5] While the paper is well-written, it lacks some contextualization in terms of prior work and potential future directions outside of the somewhat modern alignment literature that the manuscript seems to be focused on. The authors could strengthen the paper by providing a more comprehensive review of relevant work in AI interpretability as well as looking at how neuroscientists have been exploring goal-based navigation in animal brains (though there's also plenty of work looking at maze navigation in artificial agents). Work such as https://www.nature.com/articles/s41586-018-0102-6 might be a good starting point.

---

> ### Author Response · Authors · 2024-11-11
> **Response to Z5g8**
>
> Thank you for your review and your detailed feedback. We’re glad that you found the paper to be well-written and to investigate an interesting problem –- we ourselves have been quite passionate about this work!
>
> **Points of agreement.** We agree that:
> 1. The behavioral and retargetability analysis may not generalize to other settings.
> 2. We should contextualize the paper, adding a discussion section with future work and potential applications [change c2].
> 3. The literature review should include more work from neuroscience [change c3]. We will include this in the next revision.
>
> **On generalizability of our results.** In change c1, you raise concerns that the results are too narrow because we only analyzed one trained policy. While we highlighted results from the policy trained where cheese was randomly placed in the top-right $5\times 5$ corner, Appendix B.4 displays regression coefficients for $n\times n$-trained agents for $n≠5$. We also qualitatively verified that for all settings for $n>3$, subtracting the cheese vector causes the agent to ignore the cheese. These results increase the robustness and breadth of our analysis. We will better highlight these results in the revision.
>
> However, analyzing different tasks would basically require doing several additional papers’ worth of work. We are unable to complete that analysis. Furthermore, we note that [TMLR guidelines](https://jmlr.org/tmlr/editorial-policies.html) stipulate that:
>
> > The acceptance decision for a submission is based on the answers to the following questions:
> > - _Are the claims made in the submission supported by accurate, convincing and clear evidence?_
> > - _Would at least some individuals in TMLR's audience be interested in knowing the findings of this paper?_
> >
> > Papers should be accepted if they meet the criteria, _even if the contribution or significance of the work is modest._
>
> While we appreciate your concerns with change c1, we do not think that change is required in order for our work to meet TMLR’s acceptance threshold.
>
> ---
>
> Thank you again for improving our paper with your detailed feedback.

---

### Review · Reviewer_1bsk · 2024-11-11

**Summary Of Contributions:**

This paper analyzes the internal representations learned by a specific policy network to solve mazes. Authors identify certain channels to track the location of the goal. Then they modify these channels by adding interventions to partially control the policy.

**Audience:**

Yes

**Claims And Evidence:**

Yes

**Requested Changes:**

See the above weaknesses. The main concern is that there are too many manual and heuristic designs for the model and experiment parts.  More insights and analysis should be added to clarify such designs. Another point is that this paper is mainly focused on a very specific task with specifically designed models. It is difficult to see the broad usage and generalization ability to other relevant or more complex RL agents.

**Strengths And Weaknesses:**

**Strengths**
1. This paper is the first to pinpoint internal goal representations in a trained policy network.
2. Extensive experiments have been conducted to demonstrate how the activation of specific channels affects the behaviors of policy

**Weaknesses**
1. The experiments and findings are based on specific settings, models and behaviors. It is difficult to see whether the experiments can be generalized to other RL agents.

2. The most effective channels are selected by visual inspections. Are these channels can be affected by different types of mazes? Is there any insight of choosing this specific model?

3. Most of the experiment setups are hand-designed, such as the visual inspections and the choice of activations. It is difficult to understand if the behaviors are only correlated to the interventions.

---

> ### Author Response · Authors · 2024-11-18
> **Response to 1bsk**
>
> We're glad you found our experiments to be thorough and that you appreciate the novelty of the goal localization we performed! We'd like to address your concerns.
>
> > The main concern is that there are too many manual and heuristic designs for the model and experiment parts. More insights and analysis should be added to clarify such designs.
>
> It sounds like you are worried that we selected our model to be easier to study? If so, we would like to clarify that we did not make any design choices about the model -- we simply used Langosco et al.'s pretrained model. (We state this on page 1 in paragraph 4 of the introduction.)
>
> We agree that we found many features manually. Many parts of interpretability research cannot be automated (see [1] for exceptions). While we could have visually inspected the channels and then rationalized that choice as the optimum of some formal metric of e.g. equivariance, we felt it best to be honest and just say we found them by looking.
>
> However, we want our work to be as strong as possible, and appreciate your help to that end. If you still have concerns, could you specify which experiment(s) might be suspect and why?
>
> > Another point is that this paper is mainly focused on a very specific task with specifically designed models. It is difficult to see the broad usage and generalization ability to other relevant or more complex RL agents.
>
> Consider the following two points:
> 1. Due in significant part to this work and successor papers, the "cheese vector" steering technique is now broadly used for understanding and controlling LLMs. We cannot substantiate this claim without deanonymizing ourselves. If necessary, we can prove it to the AC and have them state as such to you.
>
> 2. While we share your appreciation for generalizable techniques, TMLR guidelines don't actually require "broad usage" or "generalization ability." The question is whether the scholarly contribution is interesting and rigorous:
>
> > The acceptance decision for a submission is based on the answers to the following questions:
> >
> > _Are the claims made in the submission supported by accurate, convincing and clear evidence?_
> > _Would at least some individuals in TMLR's audience be interested in knowing the findings of this paper?_
> >
> > Papers should be accepted if they meet the criteria, _even if the contribution or significance of the work is modest_.
>
> Thanks again for your feedback. We look forward to discussing further improvements with you to ensure the camera-ready is as strong as possible.
>
> [1] Conmy, Arthur, et al. "Towards automated circuit discovery for mechanistic interpretability." Advances in Neural Information Processing Systems 36 (2023): 16318-16352.

---

### Decision · Action_Editor_U4hU · 2025-01-20

**Recommendation:** Reject

**Comment:**

All three reviewers unanimously recommended rejection of the paper. While the reviewers acknowledge the authors' remarks regarding the TMLR acceptance criteria, they do not share the authors' view regarding interest to a broader audience because they consider the submitted work too "narrow".

Nevertheless, similarly as Z5g8, I do encourage the authors to improve their work, accounting for the reviewers' comments. I believe that the paper can become a valuable contribution in this way.

**Audience:**

The assessment of this aspect is not straightforward for the submitted paper. The reviewers' take on the papers is that its scope is too narrow (in terms of setting, model, and policy) to be interesting to a relevant audience and that the paper must be improved in this regard before it should be accepted. That is, there are only few insights enabling us to understand the generalizability of the presented observations as most relevant dimensions (policy, architecture, algorithms, etc.) are fixed in the conducted experiments. More concretely: after reading the paper I was left unsure regarding what to take away. But then the paper also thoroughly investigates steering in the single considered specific setup that can be conceptually similar to relevant steering tasks, e.g., in LLMs.

So where to set the bar? That is to a large extent subjective - but I also consider the paper below the bar on my reading. Thus, as this is in line with the three reviewers' perceptions, I think this criterion is not met.

**Claims And Evidence:**

The claims are well supported by evidence.

**Resubmission Of Major Revision:**

The authors may consider submitting a major revision at a later time.